# The non-canonical thioreductase Tmx2b is essential for neuronal survival during zebrafish embryonic brain development

Jordy Dekker[1], Wendy Lam[1], Herma C. van der Linde[1], Floris Ophorst[2], Charlotte de Konink[1,3,4], Rachel Schot[1], Gert-Jan Kremers[2], Leslie E. Sanderson[1], Woutje M. Berdowski[1], Geeske M. van Woerden[1,3,4], Grazia M. S. Mancini[1,*] and Tjakko J. van Ham[1,*,‡]

## ABSTRACT

Biallelic variants in thioredoxin-related transmembrane 2 protein (TMX2) can cause a malformation of brain cortical development characterized by microcephaly, polymicrogyria and pachygyria by an unknown mechanism. To investigate and visualize how TMX2 loss disrupts brain development *in vivo*, we generated zebrafish deficient for *TMX2* ortholog *tmx2b*, which during the first two developmental days showed normal brain developmental hallmarks. From 3 days onwards, however, *tmx2b* mutants had no locomotor activity; this was accompanied by cell death in the brain, but not in other organs or in the spinal cord. Strikingly, cell death in *tmx2b* mutants occurred specifically in post-mitotic neurons within a ~1.5-h timeframe, whereas neuronal progenitor and radial glial cells were preserved, and could be suppressed by inhibiting neuronal activity. *In vivo* calcium imaging showed a persistent ~2-fold increase in calcium in neurons after the onset of cell death. This suggests that calcium homeostasis underlies the *tmx2b* mutant brain phenotype. Our results indicate that TMX2 is an evolutionarily conserved, protective regulator essential specifically for post-mitotic neurons after their differentiation in the vertebrate embryonic brain.

KEY WORDS: TMX2, Zebrafish, Cortical development, Polymicrogyria, Microcephaly

## INTRODUCTION

Embryonic development of the human cerebral cortex is an intricate process directed by an interplay of molecular, genetic and environmental factors. Disruption of these factors impacts normal cortical development and can result in a malformation of cortical development with a neurodevelopmental disorder, often contributing to a global developmental delay and epilepsy with a high disease burden for the affected individual and their families (Desikan and Barkovich, 2016; Oegema et al., 2020). In the last 30 years, many genetic causes for malformations of cortical development have been identified and functional studies reveal that disruption of diverse molecular pathways, such as cell division, endoplasmic reticulum (ER) stress response, calcium homeostasis and cytoskeletal regulation, can impair cortical development (Barkovich et al., 1996; Desikan and Barkovich, 2016; Passemard et al., 2019; Severino et al., 2020; Arjun McKinney et al., 2022). This genetic heterogeneity has not only advanced our understanding of malformation of cortical development, but also significantly expanded our knowledge of the molecular pathways involved in normal cortical development of the human (Desikan and Barkovich, 2016; Severino et al., 2020).

*TMX2*, coding for thioredoxin (TRX)-related transmembrane 2 (TMX2) protein, is ubiquitously expressed in humans; however, biallelic variants in *TMX2* have been associated with severe neurodevelopmental disorders, including epileptic encephalopathy, microcephaly, and cortical malformations such as unlayered polymicrogyria and pachygyria (Vandervore et al., 2019; Ghosh et al., 2019; Meng et al., 2003). TMX2 is a member of the protein disulfide isomerase (PDI) family, consisting of over 20 ER chaperones that assist in protein folding by alternating intra- and intermolecular cysteine residues between oxidized and reduced states (Meng et al., 2003; Appenzeller-Herzog and Ellgaard, 2008; Kozlov et al., 2010; Gutierrez and Simmen, 2014; Tannous et al., 2015; Guerra and Molinari, 2020; Medinas et al., 2022). TMX2 is one of five membrane-tethered PDIs, along with TMX1, TMX3, TMX4 and TXNDC15, collectively comprising the thioredoxin-related transmembrane (TMX) protein subfamily within the PDI family (Meng et al., 2003; Guerra and Molinari, 2020). Of these proteins, only TMX5 biallelic variants have been linked to a genetic disorder, Meckel–Gruber syndrome, involving a cilia-related phenotype (Shaheen et al., 2016).

The TMX proteins share an N-terminal signal peptide required for ER targeting, a single transmembrane domain, a TRX domain containing the active site with two cysteine residues (CXXC) and a C-terminal ER retention motif (Guerra and Molinari, 2020). Unlike the other TMX proteins, TMX2 consists of multiple transmembrane domains and harbors an atypical active site, with the N-terminal cysteine being replaced by a serine residue (SNDC, instead of CXXC), and a TRX domain that is directed towards the cytosol (Vandervore et al., 2019; Oguro and Imaoka, 2019; Guerra and Molinari, 2020). As oxidoreductase activity requires a canonical CXXC motif, containing two cysteine residues, it is uncertain whether TMX2 exerts such a function (Hatahet and Ruddock, 2009; Kozlov et al., 2010; Guerra and Molinari, 2020).

Although oxidoreductase activity of TMX2 lacks definitive proof, its interaction with ER chaperones and proteins of the unfolded protein response (UPR) hint towards a putative role in protein folding by TMX2 (Vandervore et al., 2019). *In vitro* studies in mouse cortical

[1]Department of Clinical Genetics, Erasmus MC, University Medical Center Rotterdam, PO Box 2040, 3000 CA, Rotterdam, The Netherlands. [2]Department of Pathology, Optical Imaging Center, Erasmus MC University Medical Center Rotterdam, 3000 CA, Rotterdam, The Netherlands. [3]Erfelijke Neuro-Cognitieve Ontwikkelingsstoornissen, Expertise Center for Neurodevelopmental Disorders, Erasmus Medical Center, 3000 CA, Rotterdam, The Netherlands. [4]Department of Neuroscience, Erasmus Medical Center, 3000 CA, Rotterdam, The Netherlands.
*These authors contributed equally to this work

‡Author for correspondence (t.vanham@erasmusmc.nl)

L.E.S., 0000-0002-8026-406X; G.M.v.W., 0000-0003-2492-9239; T.J.v.H., 0000-0002-2175-8713

neurons and human cholangiocarcinoma cells (HuCCT1) showed that *TMX2* knockdown affects expression levels of UPR proteins (Kramer et al., 2018; Liu et al., 2023). Furthermore, TMX2 was found to localize to mitochondria-ER contacts (MERCs). Interactome analysis of TMX2 in human HEK293T cells showed several MERC-located $Ca^{2+}$ chaperones and channel proteins [e.g. calnexin (CANX), RCN2, SERCA2 (ATP2A2)] as main interactors, suggesting that TMX2 could potentially also modulate ER-mitochondria $Ca^{2+}$-regulated crosstalk, similar to TMX1 and other PDIs (Lynes et al., 2012; Raturi et al., 2016; Gutierrez and Simmen, 2018; Vandervore et al., 2019). Cultured skin fibroblasts of individuals with pathogenic variants in *TMX2* exhibited mitochondrial dysfunction, further stressing an important role for TMX2 in mitochondrial physiology (Vandervore et al., 2019). TMX2 also functions at the nuclear pore where it regulates nuclear transport via importin-β (KPNB1) (Oguro and Imaoka, 2019). Disturbance of these processes have been implicated in disruption of cortical development, hence emphasizing that the physiological function of TMX2 is essential for normal brain development (Laguesse et al., 2015; Passemard et al., 2019; Smits et al., 2023; Ding and Sepehrimanesh, 2021).

Currently, it is unknown how TMX2 loss affects cortical development *in vivo* and causes microcephaly and cortical malformation, nor how TMX2 normally regulates human brain development. The use of animal embryos has been instrumental in recapitulating steps of human brain development and, in particular, the zebrafish embryo has successfully been explored to study the effects of human pathogenic variants (Kuil et al., 2019; Berdowski et al., 2022). To obtain a better understanding of the role of TMX2 in brain development, we generated zebrafish deficient for the *TMX2* ortholog *tmx2b*. The *tmx2b*-deficient zebrafish show normal embryonic developmental hallmarks in the first two days post-fertilization (dpf). At 3 dpf, *tmx2b* knockout zebrafish show rapid-onset, massive neuronal cell death affecting most of the brain, which appears to not progress further by 5 dpf. Cell death is restricted to post-mitotic neurons as radial glia cell populations are unaffected. Calcium imaging suggested a possible dysregulation of $Ca^{2+}$ in the neurons of *tmx2b* knockout zebrafish. Our results indicate that TMX2 has a protective role in neurons by regulating $Ca^{2+}$ concentrations in the brain.

## RESULTS

### Generation of *tmx2a* and *tmx2b* knockout zebrafish

To investigate the impact of TMX2 loss *in vivo*, we generated a genetic zebrafish model by mutating both *TMX2* homologs, *tmx2a* and *tmx2b*, referred to as *tmx2a*$^{-/-}$ and *tmx2b*$^{-/-}$, by introducing frameshifting mutations in exons 1 and 3, respectively (Fig. S1). If both these mutated transcripts are not degraded by nonsense-mediated RNA decay, translation would result in a truncated protein lacking the complete catalytic TRX domain, preserving only the signal peptide and the first transmembrane domain (Fig. S2A). We observed by visual inspection that adult *tmx2a*$^{-/-}$ zebrafish did not differ from controls with respect to growth and survival, whereas early lethality after reaching the feeding stage was observed in *tmx2b*$^{-/-}$ zebrafish so only heterozygous fish could be maintained. Tmx2a and Tmx2b share, respectively, 64% and 70% protein homology with TMX2 (Fig. S2B,C). Tmx2a and Tmx2b share 67% homology and a distance tree (BLOSUM62 algorithm) showed that Tmx2b is phylogenetically more closely related to TMX2 than to Tmx2a (Fig. S2C). Furthermore, in control zebrafish brain at 5 dpf only *tmx2b* was expressed and *tmx2a* was not [average transcripts per million (TPM): *tmx2b*>53.5; *tmx2a*<0.001] and the zebrafish RNA-sequencing database (zfRegeneration) showed that across

multiple tissues and in normal embryonic/larval developmental stages *tmx2b* is the main expressed gene (Fig. S3A,B) (Nieto-Arellano and Sanchez-Iranzo, 2019). Altogether, these data show that Tmx2b is most equivalent to TMX2 and most relevant to its function in the brain; therefore, we decided to use *tmx2b*$^{-/-}$ zebrafish to model TMX2 deficiency (Fig. 1A,B).

### *tmx2b*$^{-/-}$ zebrafish embryos exhibit developmental regression from 3 dpf

As *tmx2b*$^{-/-}$ zebrafish did not reach adulthood and never survived beyond 5-10 dpf, we analyzed total body length at embryonic stages from 1 to 4 dpf as a marker of general development (Kimmel et al., 1995; Parichy et al., 2009). At 1 and 2 dpf, *tmx2b*$^{-/-}$ zebrafish embryos appeared normal and body length did not differ from that of controls (Fig. 1C,D, Fig. S4A-C). At 3 and 4 dpf, *tmx2b*$^{-/-}$ zebrafish displayed developmental decline and were shorter than *tmx2b*$^{+/+}$ and *txm2b*$^{+/-}$ zebrafish (Fig. 1C,D, Fig. S4A,D,E). Additionally, *tmx2b*$^{-/-}$ zebrafish at 3 and 4 dpf had a gray discoloration in the brain, a sign of necrotic cell death (Fig. 1E, Fig. S4F) (Byrnes et al., 2018). Interestingly, at 4 dpf the visible necrosis seemed to have been resolved in ~33% of the *tmx2b*$^{-/-}$ zebrafish, suggesting a transient phase of cell death (Fig. 1F). Heterozygous *tmx2b*$^{+/-}$ zebrafish had normal body length, did not exhibit brain necrosis and showed no visually discernable phenotype; therefore, we combined *tmx2b*$^{+/+}$ and *tmx2b*$^{+/-}$ as one control group (referred to as *tmx2b*$^{+/?}$) for the subsequent experiments (Fig. 1C-F). The first *tmx2b*$^{-/-}$ zebrafish started to develop the gray discoloration when the *tmx2b*$^{+/?}$ zebrafish siblings were between the pec-fin [60 hours post-fertilization (hpf)] and protruding-mouth (72 hpf) stages, indicating that brain necrosis initiates at the end of the embryonic phase (Fig. S4G). In addition to the brain developmental phenotype and growth delay, *tmx2b*$^{-/-}$ zebrafish also developed cardiac edema with no macroscopic evidence of involvement of other organs (Fig. 1E).

To confirm that loss of Tmx2b is responsible for the observed phenotype, we first tested whether transcript from the mutant allele undergoes nonsense-mediated mRNA decay. Sanger sequencing of mRNA isolated from fertilized oocytes of a *tmx2b*$^{+/-}$ incross revealed very low expression of the mutant allele, suggesting that mutant transcript is largely degraded (Fig. S4H). Next, to further support the suggestion that the phenotype is caused by the *tmx2b* mutation, we tested whether the brain necrosis phenotype could be rescued by injecting wild-type (WT) *tmx2b* mRNA into fertilized *tmx2b*$^{-/-}$ oocytes. Overexpression of WT *tmx2b* appeared to be toxic to both *tmx2b*$^{+/?}$ and *tmx2b*$^{-/-}$ zebrafish (Fig. S5A); however, in those embryos not affected by the toxic effects of WT *tmx2b* overexpression, we observed brains of normal appearance in 1/11 *tmx2b*$^{-/-}$ embryos (8% for 25 ng injected), 2/6 *tmx2b*$^{-/-}$ embryos (33% for 50 ng injected) and 3/6 *tmx2b*$^{-/-}$ embryos (50% for 100 ng injected) at 3 dpf, indicating that *tmx2b* mRNA can rescue the phenotype (Fig. S5B,C).

To explore further the phenotype of the *tmx2b*$^{-/-}$ zebrafish, we tested the touch response during development. At 1 and 2 dpf, *tmx2b*$^{-/-}$ mutants responded normally to touch by showing increased movement, but they were non-responsive at 3 and 4 dpf, indicating that the visually distinguishable phenotype onset in *tmx2b*$^{-/-}$ appears suddenly between 2 and 3 dpf. (Fig. 1G, Fig. S6, Movies 1-9). Next, we assessed apoptosis by visualizing Annexin A5-labeled apoptotic cells using a transgenic line used frequently detect apoptosis in zebrafish larvae (*ubb*:SecA5-mVenus$^+$ clusters; secreted Annexin A5 fused to YFP) in the brain at 2, 3 and 4 dpf (Fig. 1H,I, Fig. S7) (van Ham et al., 2010; Morsch et al., 2015). At 2 dpf, numbers of apoptotic clusters in *tmx2b*$^{-/}$ and *tmx2b*$^{+/?}$

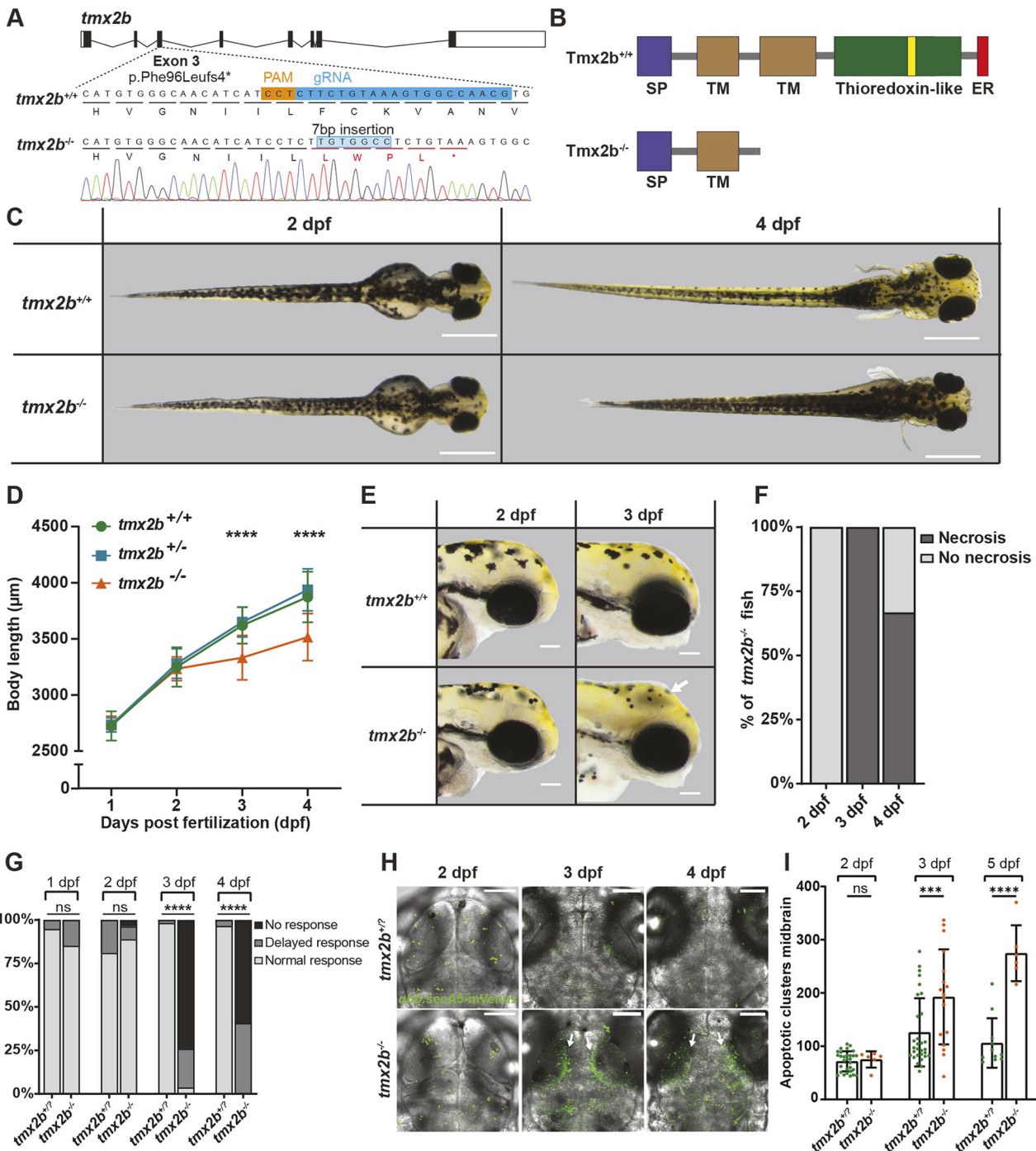

**Fig. 1. _txm2b<sup>−/−</sup>_ zebrafish display a developmental regression from 3 dpf with cell death in brain regions.** (A) Schematic of the _tmx2b_ gene (Ensembl transcript ID:ENSDART00000009858.6) with the gRNA target in exon 3. Sanger sequence data of _tmx2b<sup>−/−</sup>_ zebrafish embryo with a homozygous c.285_286insTGTGGCC, p.Phe96Leufs4* mutation. (B) Schematic of the Tmx2b protein. _tmx2b<sup>−/−</sup>_ zebrafish lack the thioredoxin-like domain containing the catalytic S-X-X-C motif (yellow box) of Tmx2b. ER, ER-retention motif; TM, transmembrane domain; SP, signal peptide sequence. (C) Representative images of same _tmx2b<sup>+/+</sup>_, _tmx2b<sup>+/−</sup>_ and _tmx2b<sup>−/−</sup>_ zebrafish at 2 and 4 days post-fertilization (dpf). _tmx2b<sup>−/−</sup>_ have normal body morphology until 2 dpf and from 3 dpf onwards display a developmental decline. Scale bars: 500 μm. (D) Body length measurements of _tmx2b<sup>+/+</sup>_, _tmx2b<sup>+/−</sup>_ and _tmx2b<sup>−/−</sup>_ zebrafish from 1 to 4 dpf. Data are represented as mean±s.d. _tmx2b<sup>+/+</sup>_, $n=9$; _tmx2b<sup>+/−</sup>_, $n=37$; _tmx2b<sup>−/−</sup>_, $n=15$. ****$P<0.0001$ (two-way ANOVA with Tukey's multiple comparisons test). (E) Representative brightfield images of _tmx2b<sup>+/+</sup>_, _tmx2b<sup>+/−</sup>_ and _tmx2b<sup>−/−</sup>_ zebrafish, lateral view of head. _tmx2b<sup>−/−</sup>_ zebrafish develop a gray discoloration in the brain region (white arrow), indicative of necrosis. This gray discoloration is not always present at 4 dpf in _tmx2b<sup>−/−</sup>_ zebrafish. Scale bars: 100 μm. (F) Quantification of the same _tmx2b<sup>−/−</sup>_ zebrafish with gray discoloration (necrosis) at 2-4 dpf. At 3 dpf, all _tmx2b<sup>−/−</sup>_ zebrafish have necrosis. At 4 dpf, 33% of the _tmx2b<sup>−/−</sup>_ zebrafish no longer have visible necrosis in the brain. $n=15$ zebrafish. (G) Quantification of _tmx2b<sup>+/?</sup>_ and _tmx2b<sup>−/−</sup>_ zebrafish with a normal/ delayed or absent touch response from 1 to 4 dpf. _tmx2b<sup>+/?</sup>_, $n=58$; _tmx2b<sup>−/−</sup>_, $n=27$ zebrafish. ****$P<0.0001$ (Fisher's exact test; delayed and no touch response groups were combined for statistical testing). (H) Representative images of _ubb_:secA5-mVenus<sup>+</sup> (green) apoptotic clusters merged with brightfield images of _tmx2b<sup>+/?</sup>_ and _tmx2b<sup>−/−</sup>_ zebrafish at 2, 3 and 4 dpf. Increased apoptosis is mostly pronounced in optic tecti (white arrows). Scale bars: 100 μm. (I) Quantification of apoptotic clusters brain of _tmx2b<sup>+/?</sup>_ and _tmx2b<sup>−/−</sup>_ zebrafish at 2, 3 and 4 dpf. Data are represented as mean±s.d. _tmx2b<sup>+/?</sup>_, $n=30,30,11$; _tmx2b<sup>−/−</sup>_, $n=6,9,6$ (2,3,4 dpf). ***$P<0.001$, ****$P<0.0001$ (two-way ANOVA, Šídák's multiple comparison test). ns, not significant ($P>0.05$).

zebrafish did not differ (Fig. 1H,I, Fig. S7). Consistent with previous findings, we observed increased numbers of apoptotic clusters in $tmx2b^{-/}$ compared to $tmx2b^{+/?}$ zebrafish at 3 and 4 dpf (Fig. 1H,I, Fig. S7). These data indicate that loss of Tmx2b does not affect general development the first 2 dpf, whereas at 3 dpf $tmx2b^{-/-}$ zebrafish all develop growth regression and massive cell death in the brain that appears to be non-progressive up to 4 dpf and affects their inability to voluntarily move after 3 dpf.

## Tmx2b loss causes neuronal cell death at 3 dpf in zebrafish embryo brains

Since we observed massive cell death in the brain, we focused on which brain cell types were affected by Tmx2 loss. Since many individuals with biallelic pathogenic loss-of-function variants in *TMX2* develop a primary microcephaly (i.e. present at birth) with abnormal cortex structure (polymicrogyria) and some show progressive microcephaly also after birth, we hypothesized that both the radial glial cells (neural and glial progenitor cells) and the generated neurons undergo cell death in $tmx2b^{-/-}$ (Desikan and Barkovich, 2016; Oegema et al., 2020; Vandervore et al., 2019; Ghosh et al., 2019). First, we determined the effect of Tmx2b loss in the *vglut2:*DsRED- and *gad1b:*GFP-labeled cells, marking largely excitatory and inhibitory neurons, respectively. At 2 dpf, the *vglut2:* DsRED- and *gad1b:*GFP-positive cell numbers were similar and did not differ between $tmx2b^{-/-}$ and $tmx2b^{+/?}$ (Fig. 2A-D, Figs S8 and S9). At 3 dpf, both *vglut2:*DsRED⁺ and *gad1b:*GFP⁺ cell loss was observed within the midbrain in $txm2b^{-/-}$ larvae (Fig. 2A-D, Figs S8 and S9). Brain size did not differ at 3 dpf between $tmx2b^{-/-}$ and $tmx2b^{+/?}$ zebrafish, indicating that the *vglut2:*DsRED⁺ and *gad1b:*GFP⁺ neurons were present in the brain at 2 dpf and died between 2 and 3 dpf (Figs S7 and S8). Neuronal cell death was not progressive as the area positive for *vglut2:*DsRED⁺ neurons increased between 3 and 5 dpf in $tmx2b^{-/-}$ zebrafish; however, the loss of *gad1b:*GFP⁺ neurons appeared to be progressive between 3 and 5 dpf (Fig. 2A-D, Figs S8 and S9).

Since the abnormal brain discoloration was only observed in the 3 dpf brain, we assessed whether neuronal loss also occurred in the neurons of the spinal cord. Both *vglut2:*DsRED- and *gad1b:*GFP-positive excitatory and inhibitory cell numbers in $tmx2b^{-/-}$ did not differ from those in $tmx2b^{+/-}$ larvae (Fig. 2E-G, Fig. S10). These data indicate that Tmx2b loss causes both excitatory and inhibitory cell death between 2 and 3 dpf limited to the brain, as the neurons in the spinal cord are unaffected.

## Glia cell populations are not directly affected by Tmx2 loss
Next, we determined whether cell death was restricted to neuronal populations or whether glial cell populations were also affected. First, we assessed the number of radial glial cells (*her4.3:*EGFP⁺ cells), which exert both astrocytic functions and neural stem cell properties in zebrafish (Mu et al., 2019; Chen et al., 2020a). Surprisingly, the radial glial cell numbers in $tmx2b^{-/}$ did not differ from those of $tmx2b^{+/?}$ control zebrafish at 2, 3 and 5 dpf (Fig. 3A,B). Although radial glial cell numbers were similar, we did observe that the soma of radial glial cells, which normally reside at the apical edge of the ventricular zone, migrated from the ventricular zone towards the midbrain/optic tectum (Fig. 3A).

Similarly, we assessed oligodendrocyte precursor cells (OPCs) by examining specific markers (*olig1:*NLS-mApple⁺ cells) and myelination levels (*mbp:*EGFP-CAAX⁺). We observed decreased *olig1:*NLS-mApple⁺ cells in $tmx2b^{-/-}$ zebrafish compared to $tmx2b^{+/?}$ at 3 and 5 dpf (Fig. 3C,D, Fig. S11). Consistent with lower *olig1:*NLS-mApple⁺ cell numbers, myelination in the midbrain/

hindbrain region was largely lacking in $tmx2b^{-/-}$ larvae (Fig. 3E,F). The observed difference in *olig1:*NLS-mApple⁺ cells appeared to be caused by a lack of proliferation between 2 and 3 dpf, as the total numbers of *olig1:*NLS-mApple⁺ cells were similar at 2 and 3 dpf in $tmx2b^{-/-}$ zebrafish (Fig. 3C,D).

Having shown effects on neurons, OPCs and myelination, we evaluated the effect of Tmx2b loss on microglia, brain-resident macrophages [Neutral Red (NR⁺) and *mpeg1:*EGFP⁺] (Colonna and Butovsky, 2017). At 3 dpf, microglia numbers were not different in $tmx2b^{-/-}$ compared to $tmx2b^{+/?}$, but they were more localized to the optic tecti regions and had an amoeboid, rounded morphology, indicative of high phagocytic activity (Fig. S12A-F) (Vidal-Itriago et al., 2022). Increased LysoTracker staining in microglia at 3 and 5 dpf was indeed observed, consistent with highly phagocytic microglia (Fig. 3G,H) (Berdowski et al., 2022). Microglia numbers were normal at 3 dpf, but at 5 dpf microglia numbers had increased ~2-fold compared to $tmx2b^{+/-}$ (Fig. S12D-F). Thus, $tmx2b^{-/-}$ microglia are likely unaffected by Tmx2b loss as their initial development is normal. Furthermore, microglia can perform their physiological functions by proliferating as a response to neuronal cell death and actively phagocytizing neuronal cell debris (Colonna and Butovsky, 2017).

## Neuronal cell death occurs rapidly at a specific time point in development
To obtain a better understanding of the onset of neuronal cell death in $tmx2b^{-/-}$ zebrafish we performed, overnight, temperature-controlled, time-lapse imaging of the DsRed⁺ and GFP⁺ neurons from 63 hpf to 75 hpf. Before onset of neuronal cell death, the $tmx2b^{-/-}$ brain developed normally and differentiated migrating neurons could be observed that were similar to those in $tmx2b^{+/?}$ zebrafish (Movies 10 and 11). Then, neurons ceased to migrate within a 10 min time interval, and a brain deformation with loss of neurons was observed, with fluorescence intensity loss as a marker for neuronal cell loss over time (Fig. 4A, Movies 10 and 11). Onset of excitatory and inhibitory cell loss coincided and occurred within a ~1.5-h timeframe (Fig. 4B). Hence, brain development appears entirely normal in $tmx2b^{-/-}$ zebrafish until the onset neuronal cell death, which occurs suddenly and spans an ~1.5 h timeframe.

## Modulating reactive oxygen species does not affect the neuronal cell death
Since TMX2 has putative oxidoreductase activity and knockdown of TMX2 has been shown to increase reactive oxygen species (ROS) in cholangiocarcinoma cells *in vitro*, we reasoned that increased cellular ROS potentially could cause neuronal cell death (Kannan and Jain, 2000; Liu et al., 2023). Therefore, we tested the general antioxidant *N*-acetylcysteine (NAC) and two inhibitors of $H_2O_2$-synthesizing proteins in the ER (EN460: ERO1 inhibitor; GKT137831: NOX4 inhibitor) (Table S3) (Chen et al., 2017; Geldenhuys et al., 2017; Yoboue et al., 2018; Pedre et al., 2021). None of the drugs ameliorated the $tmx2b^{-/-}$ zebrafish phenotype, as all treated zebrafish still had growth delay and visible neuronal cell death (Fig. S13). To investigate further whether ROS contributed to the $tmx2b^{-/-}$ phenotype, we increased cellular ROS by $H_2O_2$ treatment. Surprisingly, a decrease in visible brain necrosis was observed in $tmx2b^{-/-}$ $H_2O_2$-treated zebrafish at 3 dpf (Fig. S14A-C). However, $H_2O_2$ treatment also resulted in a minor developmental delay in $H_2O_2$-treated $tmx2b^{+/-}$, explaining the decrease in brain necrosis (Fig. S14B). Nonetheless, $H_2O_2$ treatment did not result in an earlier onset of the necrotic phenotype compared to untreated $tmx2b^{-/-}$

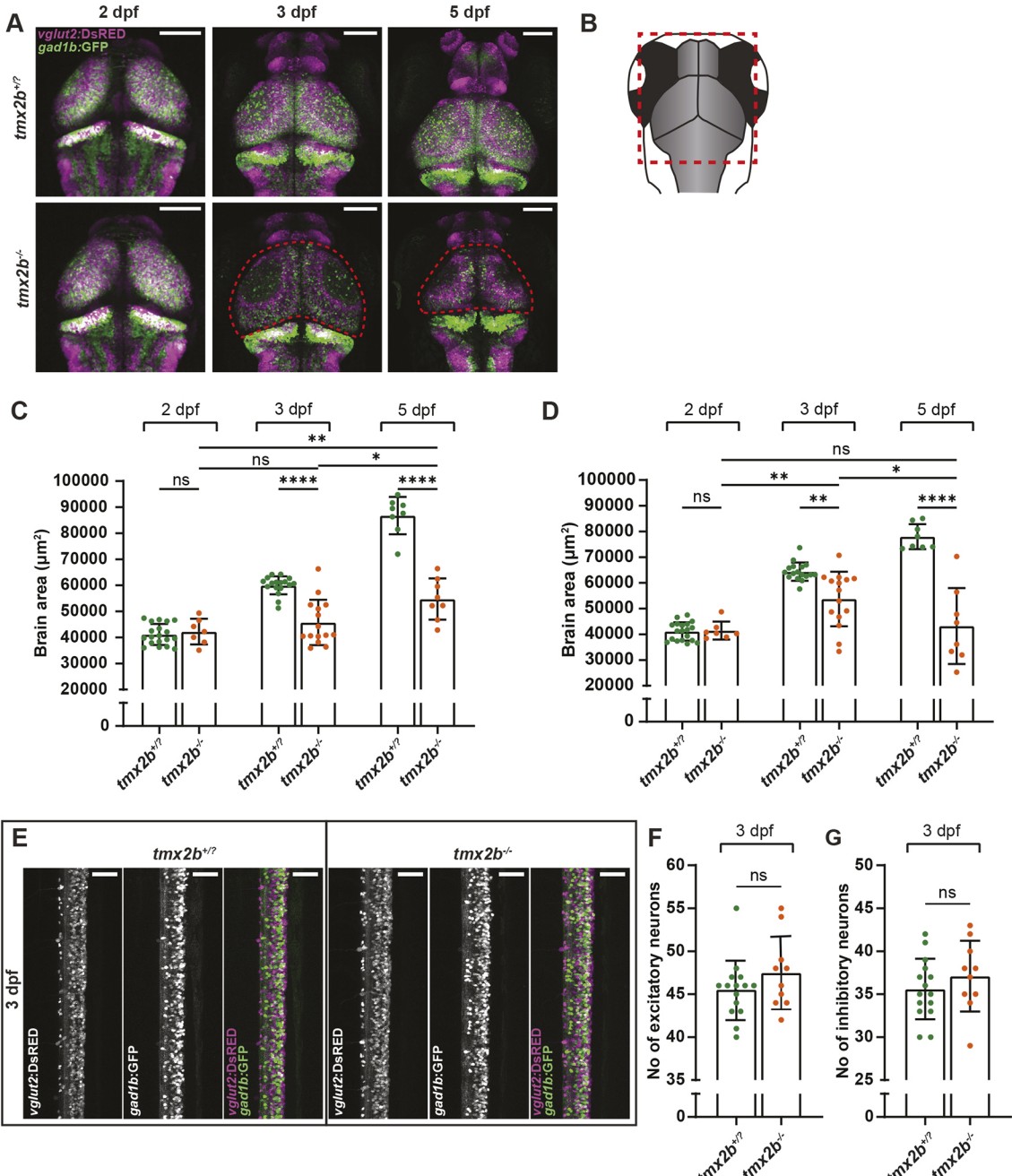

**Fig. 2. Neurons in central brain regions in *tmx2b*⁻/⁻ zebrafish undergo cell death at 3 dpf.** (A) Representative images of *vglut2*:DsRED⁺ excitatory neurons (magenta) and *gad1b*:GFP⁺ inhibitory neurons (green) in *tmx2b*⁺/� and *tmx2b*⁻/⁻ zebrafish brains at 2, 3 and 5 dpf. Red dashed lines indicate areas of neuronal cell loss. Scale bars: 100 μm. (B) Schematic representation of the zebrafish brain indicating the region selected for imaging. (C) Quantification of excitatory neuron brain area in *tmx2b*⁺/ᵈ and *tmx2b*⁻/⁻ zebrafish at 2, 3 and 5 dpf. Excitatory neuronal cell death is observed at 3 dpf in *tmx2b*⁻/⁻ zebrafish, but a minor recovery is observed at 5 dpf. (C) Quantification of inhibitory neuron brain area in *tmx2b*⁺/ᵈ and *tmx2b*⁻/⁻ zebrafish at 2, 3 and 5 dpf. Contrary to the excitatory neurons, inhibitory neuronal loss is progressive from 3 dpf onwards. (A-C) *tmx2b*⁺/ᵈ, *n*=19,17,8; *tmx2b*⁻/⁻, *n*=7,15,8 (2,3,5 dpf). *$P<0.05$, **$P<0.01$, ***$P<0.001$, ****$P<0.0001$ (two-way ANOVA, Tukey's multiple comparisons test). (E) Representative images of *vglut2*:DsRED⁺ excitatory neurons and *gad1b*:GFP⁺ inhibitory neurons in spinal cord of *tmx2b*⁺/ᵈ and *tmx2b*⁻/⁻ zebrafish at 3 dpf. Scale bars: 50 μm. (F) Quantification of numbers of excitatory neurons in spinal cord of in *tmx2b*⁺/ᵈ and *tmx2b*⁻/⁻ zebrafish at 3 dpf. (G) Quantification of numbers of inhibitory neurons in spinal cord of in *tmx2b*⁺/ᵈ and *tmx2b*⁻/⁻ zebrafish at 3 dpf. (E-G) *tmx2b*⁺/ᵈ, *n*=15; *tmx2b*⁻/⁻, *n*=10 zebrafish. One-way ANOVA with Šídák's multiple comparisons test. Data are represented as mean±s.d. ns, not significant ($P>0.05$).

zebrafish, suggesting that loss of neurons in *tmx2b*⁻/⁻ is not primarily driven by increased ROS.

## Ca²⁺ dysregulation in *tmx2b*⁻/⁻ zebrafish brain

Individuals with pathogenic *TMX2* variants often present with severe epilepsy (Vandervore et al., 2019; Ghosh et al., 2019). It is

unknown whether any epilepsy-induced neuronal cell damage and eventual death relates to the progressive disease course observed in these individuals (Vandervore et al., 2019; Ghosh et al., 2019). Possibly, abnormal neuronal firing could contribute to neuronal cell death, and to test this hypothesis we treated zebrafish with the voltage-gated sodium channel blocker tricaine (Carter et al., 2011;

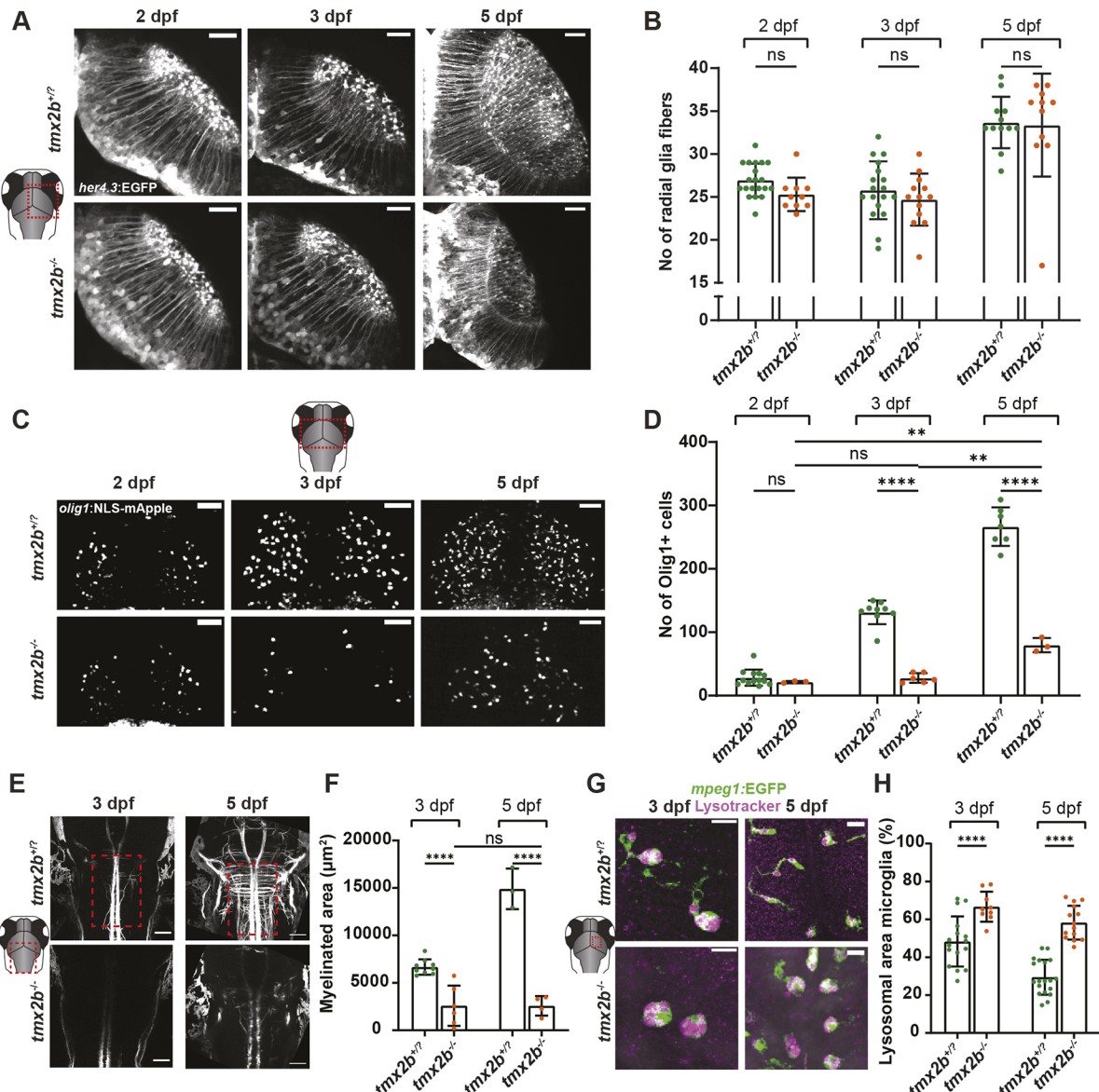

**Fig. 3. Glial cell populations in *tmx2b⁻/⁻* zebrafish.** (A) Representative images of *her4.3*:EGFP⁺ radial glia cells in right midbrain hemisphere of *tmx2b⁺/ᐟ* and *tmx2b⁻/⁻* zebrafish at 2, 3 and 5 dpf. Scale bars: 25 µm. (B) Quantification of the number of radial glial fibers in the right midbrain hemisphere of *tmx2b⁺/ᐟ* and *tmx2b⁻/⁻* zebrafish at 2, 3 and 5 dpf. *tmx2b⁺/ᐟ*, n=20,18,12; *tmx2b⁻/⁻*, n=11,13,11 (2,3,5 dpf). Two-way ANOVA, Šídák's multiple comparisons test. (C) Representative images of *olig1*:NLS-mApple (magenta) OPCs in the midbrain of *tmx2b⁺/ᐟ* and *tmx2b⁻/⁻* zebrafish at 2, 3 and 5 dpf. Scale bars: 100 µm. (D) Quantification of OPCs in midbrain of *tmx2b⁺/ᐟ* and *tmx2b⁻/⁻* zebrafish at 2, 3 and 5 dpf. *tmx2b⁺/ᐟ*, n=13,9,7; *tmx2b⁻/⁻*, n=3,6,3 (2,3,5 dpf). **$P<0.01$, ****$P<0.0001$ (two-way ANOVA, Tukey's multiple comparisons test). (E) Representative images of *mbp*:EGFP-CAAX in the hindbrain region of *tmx2b⁺/ᐟ* and *tmx2b⁻/⁻* zebrafish at 3 and 5 dpf. Scale bars: 50 µm. (F) Quantification of myelinated area in the hindbrain of *tmx2b⁺/ᐟ* and *tmx2b⁻/⁻* zebrafish at 3 and 5 dpf. Measured area is indicated by the red dashed box in E. (E,F) N=1,1 experiments (3,5 dpf); *tmx2b⁺/ᐟ*, n=9,6; *tmx2b⁻/⁻*, n=6,4 (3,5 dpf). ****$P<0.0001$ (two-way ANOVA, Tukey's multiple comparisons test). (G) representative images of *mpeg1*:GFP⁺ microglia (green) and LysoTracker in the midbrain of *tmx2b⁺/ᐟ* and *tmx2b⁻/⁻* zebrafish at 3 and 5 dpf. Microglia in *tmx2b⁻/⁻* zebrafish have an amoeboid morphology and higher lysosome concentration. Scale bars: 15 µm. (H) Quantification of lysosomal area within microglia in the midbrain of *tmx2b⁺/ᐟ* and *tmx2b⁻/⁻* zebrafish at 3 and 5 dpf. Each dot represents the average value of six microglia from a single zebrafish brain. *tmx2b⁺/ᐟ*, n=16,19 (3,5 dpf); *tmx2b⁻/⁻*, n=9,13 zebrafish (3,5 dpf). ****$P<0.0001$ (two-way ANOVA with Šídák's multiple comparisons test). Data are represented as mean±s.d. ns, not significant ($P>0.05$).

Attili and Hughes, 2014). Strikingly, tricaine treatment prevented necrosis development in ∼66% of *tmx2b⁻/⁻* zebrafish at 3 dpf (Fig. 5A-C). In contrast to untreated zebrafish, body length of tricaine-treated *tmx2b⁻/⁻* zebrafish was not different from that of treated and untreated control zebrafish; therefore, decreased necrosis could not be explained by a general developmental delay of the zebrafish (Fig. 5B). If inhibition of neuronal action potentials by the anesthetic tricaine prevented necrosis, induction of neuronal activity or eliciting seizures could speed up the onset of necrosis. We

utilized 4-aminopyridine (4-AP), a potassium channel antagonist, to induce over-excitation of neurons in the *tmx2b⁻/⁻* zebrafish, which did not show an earlier onset of neuronal cell death (Fig. S14D-F) (Ellis et al., 2012; Perenthaler et al., 2020). At 2 dpf, all 4-AP-treated *tmx2b⁻/⁻* zebrafish showed normal development and displayed no gray discoloration in the brain, whereas at 3 dpf extensive necrosis, similar to that observed in the untreated group in *txm2b⁻/⁻* larvae, was observed (Fig. S14D-F). Altogether, these data indicate that suppressing neuronal excitation rescues neuronal cell death, whereas

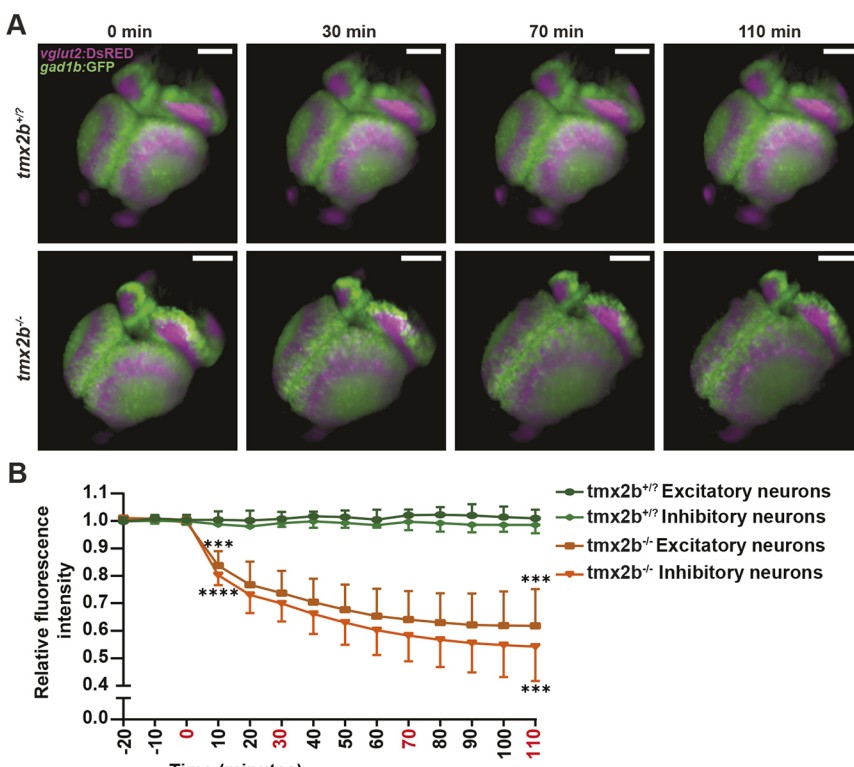

**Fig. 4. Neuronal cell death has a rapid onset.** (A) 3D reconstructions of *vglut2*:DsRED⁺ excitatory neurons (magenta) and *gad1b*:GFP⁺ inhibitory neurons (green) of $tmx2b^{+/-}$ and $tmx2b^{-/-}$. Zebrafish were imaged from 58 hpf to 75 hpf. This figure only shows the images around onset of the neuronal cell death in $tmx2b^{-/-}$. (B) Quantification of relative fluorescence intensity of excitatory and inhibitory neurons in the midbrain as a measurement for neuronal cell loss. The neuronal cell loss starts within a 10-min period and the majority of cell loss occurs within ~1.5-h time window. *n*=3 zebrafish for both groups. \*\*\**P*<0.001, \*\*\*\**P*<0.0001 (multiple *t*-tests with Holm–Šidák's multiple comparison correction). Data are represented as mean ±s.d. Asterisks denoting significance are only shown for excitatory neurons (upper asterisks) and inhibitory neurons (lower asterisks) at time points 0 and 110 min. However, all time points ≥0 min reached significance for both excitatory and inhibitory neurons.

stimulating neuronal activity/seizures cannot accelerate the onset of neuronal cell death phenotype in $tmx2b^{-/-}$ zebrafish. Alternatively, the effect of tricaine may be unrelated to seizure suppression and rather related to an additional function of voltage-gated sodium channels in microglia activity (Craner et al., 2005).

Besides oxidoreductase activity, various PDIs also regulate $Ca^{2+}$ flow between the ER and mitochondria (Gutierrez and Simmen, 2018). Since TMX2 can localize to MERCs and can interact with calcium-binding proteins (Vandervore et al., 2019), we tested the hypothesis that $Ca^{2+}$ homeostasis is dysregulated in $tmx2b^{-/-}$ zebrafish (Lynes et al., 2012; Vandervore et al., 2019). We employed the neuronal $Ca^{2+}$ sensor NLS-GCaMP6 s (*elavl3*:NLS-GCaMP6s) in tricaine-anesthetized zebrafish, thereby preventing any influence of neuronal activity, to assess baseline nuclear and cytosolic $Ca^{2+}$ concentrations (Al-Mohanna et al., 1994; Forster et al., 2017). Already at 2 dpf, before the onset of cell death, we observed that nuclear and cytoplasmic $Ca^{2+}$ concentrations in neurons were slightly decreased in $tmx2b^{-/-}$ zebrafish compared to $tmx2b^{+/?}$ (Fig. 5D,E, Fig. S15). In contrast, at 3 dpf neuronal $Ca^{2+}$ concentration was increased by ~2-fold in $tmx2b^{-/-}$ zebrafish (Fig. 5D,E, Fig. S15). Our previous data could not identify abnormalities during the first 2 days of development in $tmx2b^{-/-}$ zebrafish; however, these data indicate that $Ca^{2+}$ is reduced at a stage when we do not observe other cellular phenotypes, thus preceding major loss of neurons due to apoptosis. This suggests that diminished cytoplasmic calcium precedes neuronal cell death.

## DISCUSSION

Advanced genomic analysis has enabled identification of human brain disorders caused by genetic mutations in previously unknown genes at high pace. Hence, elucidation of the physiological gene function and disease mechanism follows identification of gene mutations and often requires recapitulation of the disorder in animal models. Here, we established a zebrafish disease model for

TMX2-related malformation of cortical development and neurological disease and show that TMX2 is essential for survival of post-mitotic neurons. We identified Tmx2b as the *TMX2* ortholog and found that Tmx2b inactivation interferes with zebrafish brain development. Both excitatory and inhibitory neurons in Tmx2b-deficient zebrafish underwent cell death at the end of the zebrafish embryonic phase (Kimmel et al., 1995). This cell death had a sudden onset and occurred within a 1.5-h timeframe and was not progressive until 5 dpf, beyond which age the embryos did not survive. Cell death occurred specifically in neurons, while OPCs were decreased in number without observable cell loss and myelination was largely absent in the brain. Neuronal progenitors/radial glial cells and microglia were not directly compromised by Tmx2b loss. Lastly, we showed that neuronal cell death could be suppressed by a voltage-gated sodium channel blocker and that $Ca^{2+}$ dysregulation preceded neuronal loss, consistent with a role of TMX2 in regulating $Ca^{2+}$ homeostasis in post-mitotic neurons.

Individuals with biallelic pathogenic variants in *TMX2* present with severe neurodevelopmental disorder, epilepsy, primary and/or progressive microcephaly and polymicrogyria (Vandervore et al., 2019; Ghosh et al., 2019). Primary microcephaly (i.e. microcephaly present at birth) results from either too little proliferation or increased cell death/apoptosis of the neural progenitor cells/radial glia cells (Desikan and Barkovich, 2016; Pirozzi et al., 2018). Some individuals with *TMX2*-related brain malformation show progression of the microcephaly through infancy, which indicates a progressive interference with brain development beyond an early proliferative stage. Our previous studies suggested that TMX2 loss results in cell death of the neural progenitor population (Ghosh et al., 2019). This study in zebrafish instead shows that, not the neural progenitor cells, but the post-mitotic excitatory and inhibitory neurons undergo cell death at a specific point during brain development. The cell death of early born neurons observed in our zebrafish model could explain the microcephaly observed in individuals with pathogenic variants in

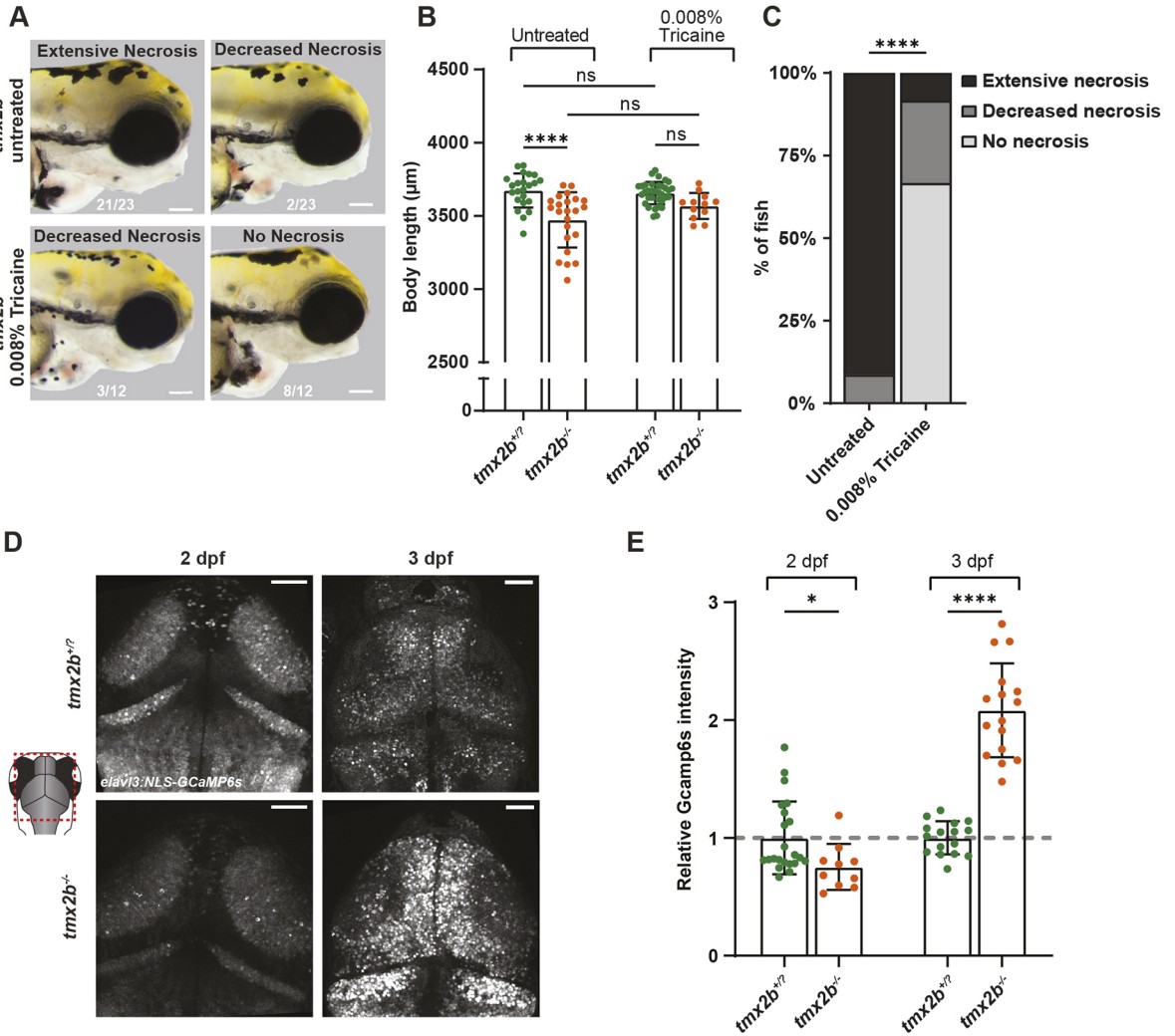

**Fig. 5. Ca²⁺ is dysregulated in neurons of _tmx2b⁻/⁻_ zebrafish.** (A) Brightfield images of the lateral view of the head of 3 dpf zebrafish untreated (top) and 0.008% tricaine treated (bottom). Of the 0.008% tricaine-treated _tmx2b⁻/⁻_ zebrafish, >50% exhibited no necrosis at 3 dpf. Scale bars: 100 μm. Numbers at the bottom indicate counts of zebrafish with the specified genotype and phenotype. (B) Body length measurements of 3 dpf untreated and 0.008% tricaine-treated _tmx2b⁺/?_ and _tmx2b⁻/⁻_ zebrafish. ****$P<0.0001$ (two-way ANOVA with Tukey's multiple comparisons test). (C) Quantification of the necrosis phenotypes shown in A. ****$P<0.0001$ (Fisher's exact test; extensive and decreased necrosis groups were combined for statistical testing). (A-C) _tmx2b⁺/?_, $n=23,34$; _tmx2b⁻/⁻_, $n=23,12$ zebrafish (untreated, 0.008% tricaine treated). (D) Representative images of elavl3:NLS-GCaMP6s neurons in _tmx2b⁺/?_ and _tmx2b⁻/⁻_ zebrafish brain at 2 and 3 dpf. Scale bars: 50 μm. (E) Relative NLS-GCaMP6s fluorescence intensity normalized to the average _tmx2b⁺/?_ value at 2 or 3 dpf. _tmx2b⁺/?_, $n=22,16$ (2,3 dpf); _tmx2b⁻/⁻_, $n=10,16$ (2,3 dpf). *$P<0.05$, ****$P<0.0001$ (two-way ANOVA with Šídák's multiple comparisons test on non-normalized data). Data are represented as mean±s.d. ns, not significant ($P>0.05$).

_TMX2_, and how the neuronal damage could progress also after the initial proliferation stage.

The brain of newborns with _TMX2_ loss often shows polymicrogyria, which is a defect of cortical organization that is sometimes developmental, sometimes infectious, as in, for example, congenital cytomegalovirus (CMV) infection (Vandervore et al., 2019). Brain MRI and pathology of affected individuals show that the polymicrogyria is unlayered, which is a sign of early disruption of cortical development between gestational weeks 13 and 16, very similar to the CMV infection (Barth, 1987; Jansen and Andermann, 2005; Vandervore et al., 2019). In human brains affected by CMV infection, foci of necrosis close to the polymicrogyria region have been detected, suggesting that localized cell death is linked to the polymicrogyria (Jansen and Andermann, 2005; Hawkins-Villarreal et al., 2023). We hypothesize that the necrotic and apoptotic neurons during brain development in the _tmx2b⁻/⁻_ zebrafish recapitulate events leading to the cortical malformation seen on MRI and pathology of

individuals with pathogenic _TMX2_ variants. Our findings and conclusions also support the hypothesis that a damaging process resulting in neuronal cell death in the cortical plate causes the malformation of cortical development.

The phenotype of _tmx2b⁻/⁻_ zebrafish appears more severe than what is typically observed in individuals with pathogenic _TMX2_ variants (Vandervore et al., 2019; Ghosh et al., 2019). While some affected individuals with pathogenic _TMX2_ variants do not survive beyond the first month of life, others can reach adulthood, unlike the _tmx2b⁻/⁻_ zebrafish (Vandervore et al., 2019). Although _TMX2_ pathogenic variants are supposed to have a loss-of-function effect, complete absence of TMX2 has not been proven in all affected individuals with biallelic variants (Vandervore et al., 2019; Ghosh et al., 2019). Most affected individuals present either with biallelic missense variants or with compound heterozygous truncating variant, i.e. null mutation, and a missense variant. _Tmx2_ knockout mice are embryonically lethal and our _tmx2b⁻/⁻_ zebrafish also do

not survive after reaching the feeding stage (Mager, 2019). This suggests that a complete loss of *TMX2* is lethal during embryonic development and could explain why no individuals have been described until now with biallelic truncating variants.

Although *in vitro* studies indicated that loss of TMX2 increases cellular ROS, subjecting zebrafish larvae to small molecules including a general antioxidant and two inhibitors of $H_2O_2$-synthesizing proteins in the ER did not appear to modulate the phenotype of $tmx2b^{-/-}$ zebrafish (Liu et al., 2023). We cannot exclude the possibility, however, that increased ROS could be partly responsible for the phenotype. Another possible explanation for the neuronal cell death is that Tmx2b loss results in a severe status epilepticus, which is known to be detrimental for neurons (Dingledine et al., 2014). Anesthetizing our $tmx2b^{-/-}$ zebrafish did not prevent neuronal cell death onset. However, drug-induced epilepsy did not result in an earlier onset of the phenotype, suggesting that prolonged and excessive neuronal activity is not the only cause of the $tmx2b^{-/-}$ cell death, but could still be its consequence. Our results, however, point towards a general $Ca^{2+}$ dysregulation in neurons. $Ca^{2+}$ concentrations in the nucleus, which are indirectly a measurement for $Ca^{2+}$ in the cytosol, were decreased at 2 dpf and increased at 3 dpf in $tmx2b^{-/-}$ zebrafish (al-Mohanna et al., 1994). Tricaine anesthesia diminishes neuronal action potentials, consequently inhibiting $Ca^{2+}$ influx in neurons, which could counteract the $Ca^{2+}$ dyshomeostasis in $tmx2b^{-/-}$ zebrafish brain.

Intracellular $Ca^{2+}$ functions as a second messenger in various cellular processes, including neurotransmitter release and the ER serves as the largest $Ca^{2+}$-storing organelle (Prins and Michalak, 2011; Sudhof, 2012). The ER contains various $Ca^{2+}$ channels that transport $Ca^{2+}$ into the cytosol and subsequently towards the mitochondria. $Ca^{2+}$ transport from the ER towards the mitochondria increases oxidative phosphorylation, but prolonged $Ca^{2+}$ flux can also induce apoptosis (Gutierrez and Simmen, 2018). $Ca^{2+}$ is released from the ER via ryanodine receptors (RyRs) and inositol 1,4,5-trisphosphate receptors (IP3Rs) of which the IP3R1 and RyR3 are primarily utilized in the brain (del Prete et al., 2014; Arruda and Hotamisligil, 2015). Loss-of-function variants of *ITPR3*, encoding IP3R1, are associated with spinocerebellar ataxia 15 and knockout mice also display a cerebellar ataxia phenotype (van de Leemput et al., 2007). Since we observed a brain-specific effect of Tmx2b loss, without cerebellar anomalies, it seems unlikely that loss of TMX2 directly impairs the function of IP3R1. $Ca^{2+}$ uptake from the cytosol towards the ER is executed by ATP2A2 (also known as SERCA2), which is known interactor of TMX2 (Vandervore et al., 2019). Heterozygous loss-of-function variants in *ATP2A2* cause Darier disease, which is primarily a skin disorder; however, neuropsychiatric diseases, such as bipolar disorder and schizophrenia, are also associated with this disease (Sakuntabhai et al., 1999; Gordon-Smith et al., 2018). Interestingly, brain-specific *Atp2a2* knockout in mice is embryonically lethal and these mice display destructive intracerebral hemorrhages (Nakajima et al., 2021). Whether these intracerebral hemorrhages were the result of a primary vascular defect or secondary to neuronal cell death was not determined (Nakajima et al., 2021). Lack of physiological interaction with $Ca^{2+}$-regulating proteins could therefore be the cause of neuronal cell death in the absence of Tmx2b (Joshi et al., 2023; Kolobkova et al., 2017).

A previous study showed that treatment of zebrafish with either rotenone and azide – a mitochondrial complex I and IV inhibitor, respectively – also induced cell death exclusively in the brain, reminiscent of our observations in the $tmx2b^{-/-}$ zebrafish (Byrnes et al., 2018). Fibroblasts of individuals with pathogenic

variants of *TMX2* also display mitochondrial dysfunction and exhibit suppressed mitochondrial respiration after treatment with the mitochondrial uncoupler FCCP, which reflects reduced reserve capacity (Vandervore et al., 2019). Additionally, all tested patient-derived cultured fibroblasts showed exhibited decreased rotenone-dependent respiration, which indicates reduced activity of complex I (Vandervore et al., 2019). Altogether, these data point towards a potential mitochondrial impairment in the $tmx2b^{-/-}$ zebrafish. As mitochondrial respiration is regulated by $Ca^{2+}$ flux from the ER to mitochondria via MERCs and TMX2 is localized at these contact points, loss of TMX2 could compromise this $Ca^{2+}$ flow, consequently impairing mitochondrial function. This mitochondrial dysfunction results in neuronal cell death specifically, due to the relatively high energy demand of post-mitotic neurons, required for migration, axonal and dendritic growth, and synapse formation.

The question remains as to why decreased cytosolic $Ca^{2+}$ concentration is observed in *tmx2*-deficient zebrafish at 2 dpf, followed by an increased concentration at 3 dpf. A possible explanation comes from a recent preprint demonstrating that TMX2 has a mitochondria-ER tethering function which decreases the distance between the ER and mitochondria, thereby decreasing $Ca^{2+}$ flux from ER to mitochondria (Chen et al., 2024 preprint). In the absence of TMX2, an increased $Ca^{2+}$ flux is observed from the ER to mitochondria. It is possible that the decreased cytosolic $Ca^{2+}$ in the $tmx2b^{-/-}$ zebrafish at 2 dpf is the result of an increased $Ca^{2+}$ concentration in the mitochondria, leading to calcium depletion in other organelles and the cytoplasm. This $Ca^{2+}$ overload can subsequently result in swelling of the mitochondria with an eventual rupture of the outer membrane, leading to $Ca^{2+}$ release into the cytoplasm, which could explain the increased cytoplasmic $Ca^{2+}$ in $tmx2b^{-/-}$ zebrafish brain at 3 dpf (Giorgi et al., 2012).

Although TMX2 is ubiquitously expressed, we only observed a largely neuron-specific effect in $tmx2b^{-/-}$ zebrafish, and a reduction of OPCs and myelination (Meng et al., 2003). Between 2 and 3 dpf, OPC numbers remained the same in $tmx2b^{-/-}$ zebrafish, suggesting a lack of proliferation and not necessarily cell death of OPCs. It is well established that OPC proliferation and myelination is dependent on neuronal activity and the presence of axons (Barres and Raff, 1993; Almeida et al., 2011; Hines et al., 2015; Marisca et al., 2020). Therefore, the decreased OPC numbers and lack of myelination is either a secondary effect to the neuronal cell death, or the result of insufficient differentiation of common radial glial progenitors into OPCs. We did not detect extensive cell death in organs other than the brain at 3-5 dpf. The only abnormalities we observed outside of the brain in $tmxb2^{-/-}$ zebrafish were cardiac edema and a minor decrease in longitudinal growth, which did not precede neuronal cell death and are therefore likely secondary to the brain developmental phenotype. Hence, cell death caused by Tmx2b loss appears to be limited to neurons. Mitochondrial dysfunction in $tmx2b^{-/-}$ in combination with a higher energy demand of neurons could also provide a possible explanation for the neuronal cell death. $Ca^{2+}$ signaling and currents differ among neuronal and glial cell types. Therefore, neuronal cell death in the brain of $tmx2b^{-/-}$ zebrafish could be related to how $Ca^{2+}$ signaling is organized in the different cell types throughout the central nervous system (Alves et al., 2019).

The current study focused on experiments in germline *tmx2b* knockouts. Inducible, targeted knockout approaches could determine whether *tmx2b* is specifically required for neuronal survival between 2 and 3 dpf or for which specific cell types Tmx2b is required (Li et al., 2020). Another limitation of our study is that we are not certain whether and to what extent ROS were reduced by small molecule treatment. We tried to visualize cellular ROS with

2′,7′-dichlorofluorescein diacetate (DCFH-DA; Sigma-Aldrich, D6883), which has been used previously *in vivo* in zebrafish (Kishi et al., 2008; Chen et al., 2020b). We noticed that the tissue penetrance of DCFH-DA *in vivo* was very limited, and staining was mainly observed in either the larval gut or blood vessels, similar to previous studies (Kishi et al., 2008; Chen et al., 2020b). In one study, ROS punctae were detected with DCFH-DA in the spinal cord region at 3.5 dpf, but we were unable to replicate this observation (Kishi et al., 2008).

In conclusion, we established, using a zebrafish genetic disease model, that Tmx2b is essential for survival of specifically post-mitotic neurons. Loss of Tmx2b does not affect initial brain and neuronal development; however, at the end of the embryonic developmental phase excitatory and inhibitory neurons undergo cell death within a 1.5-h timeframe. The cause of the associated decrease of the normal oligodendrocyte population and insufficient myelination needs to be further explored. Radial glial cell and microglial populations are not affected by Tmx2b loss. Based on these observations, we hypothesize that TMX2 is essential for the survival of post-mitotic neurons during brain development. Its loss causes neuronal dysregulation of $Ca^{2+}$, which, through impaired mitochondrial function, could cause cell death and depletion of the post-mitotic neuron population, providing a model for the microcephaly and disruptive polymicrogyria observed in individuals with *TMX2* mutations.

## MATERIALS AND METHODS
### Zebrafish housing and husbandry
Zebrafish were under standard housing and husbandry conditions (Alestrom et al., 2020). The adult animals were fed twice a day on a 14 h-10 h light-dark cycle. Zebrafish embryos and larvae were kept at 28°C in E3 medium buffered with 20 mM HEPES (pH 7.2) (referred to as E3) on a 14 h-10 h light-dark cycle until 5 dpf. To prevent pigmentation, E3 medium was changed to 0.003% (m/v) 1-phenyl 2-thiourea (PTU; Sigma-Aldrich) at 24 hpf. Fifty zebrafish embryos/larvae were kept in a Petri dish containing 25 ml E3 medium. The transgenic zebrafish lines used in this study are listed in Table S1 (Yeo et al., 2007; van Ham et al., 2010; Almeida et al., 2011; Ellett et al., 2011; Koyama et al., 2011; Satou et al., 2013; Forster et al., 2017; Marisca et al., 2020). The experiments at a defined dpf in this study were always performed in the afternoon between 14:00 and 17:00 h, so an experiment performed on a 3 dpf zebrafish larva was performed on an embryo 72-75 hpf of age.

### CRISPR-Cas9 genome editing
Alt-R® CRISPR-Cas9 crRNAs were designed and ordered via the integrated DNA technologies (IDT) website. The crRNA sequence targeted against exon 1 of *tmx2a* is 5′-GGAGTCTCCGTCCTCTCTCT-3′ and exon 3 of *tmx2b* 5′-CGTTGGCCACTTTACAGAAG-3′. Equal volumes of Alt-R® CRISPR-Cas9 tracrRNA and crRNA in duplex buffer were incubated at 95°C for 5 min, followed by cooling down at room temperature (RT) to allow cr:tracrRNA complex formation. The Sp-Cas9 plasmid (Addgene plasmid #62731) was utilized to synthesize SpCas9 as described previously (D'Astolfo et al., 2015). For the formation of cr:tracrRNA-Cas9 ribonucleoproteins, 25 pmol SpCas9 was mixed with 50 pmol cr:tracrRNA and incubated at RT for 5 min. Subsequently, 3 µl 300 mM KCl and 0.3 µl Phenol Red were added, and 1 nl of this mixture was injected into one-cell-stage WT zebrafish embryos. Indel frequency was determined as previously described (Brinkman et al., 2014). Primers used are listed in Table S2. Founder zebrafish positive for indels in *tmx2a* and *tmx2b* respectively were outcrossed against WT zebrafish to generate a heterozygous F1 generation. Genotypes of F1 adult zebrafish were determined by Sanger sequencing. Zebrafish heterozygous for a 1 bp insertion in *tmx2a* and a 7 bp insertion in *tmx2b* were selected. *Tmx2b* mutant zebrafish are available upon request.

### Genotyping zebrafish embryos *tmx2b* by allele-specific PCR
Since homozygous loss of *tmx2b* was embryonically lethal, we performed all experiments on an incross of heterozygous ($tmx2b^{+/-}$) zebrafish. WT and

heterozygous zebrafish siblings ($tmx2b^{+/?}$) served as the control group for $tmx2b^{-/-}$. After each experiment, zebrafish embryos and larvae were euthanized and DNA extraction was performed by lysis of the embryos in 40 µl 50 mM NaOH, followed by incubation at 95°C for 30 min. Samples were cooled down to RT and 4 µl 1 M Tris-HCl pH 8.0 was added. Allele-specific PCR consisted of three different primers, a forward and reverse primer and within the middle an allele-specific primer able to bind either only the WT or mutant allele (Table S2). Allele-specific PCR with FastStart Taq (Roche Diagnostics) was performed under following conditions: 95°C for 5 min, 10 cycles of 94°C for 30 s, touchdown 65→60°C (−0.5°C/cycle) for 30 s, 72°C for 45 s, followed by 25 cycles of 94°C for 30 s, 60°C 30 s, 72°C for 45 s and finished by a final step of 72°C for 5 min. 3 µl PCR product was visualized on 2% agarose gel in Tris-acetate-EDTA buffer.

### Touch response analysis
Zebrafish embryos were placed individually into 48-well plates containing 1 ml E3 medium. Touch response was provoked with a plastic loading pipette tip at 1 dpf. At 2 dpf, zebrafish embryos were first dechorionated and touch response was provoked with a 23 G needle from 2 to 4 dpf. Touch responses were provoked a maximum of three times per zebrafish and recorded as: (1) normal (upon first touch zebrafish swims out of view), (2) delayed (>1 provocations before effective swimming or ineffective swimming response, as observed by only twitching of the body), or (3) no response (after three provocations no movement visible). Movies were recorded with an Olympus SZX116 microscope with a DP72 camera (Olympus).

### Neutral Red staining
Zebrafish larvae at 3 and 5 dpf were incubated in 2.5 µg/ml Neutral Red (Sigma-Aldrich) dissolved in E3 containing 0.003% (m/v) PTU for 2 h at 28°C. After incubation, zebrafish were washed and incubated in E3 containing 0.003% (m/v) PTU for 20 min at 28°C. Stained zebrafish larvae were anesthetized with 0.016% (m/v) ethyl 3-aminobenzoate methanesulfonate salt (tricaine; Sigma-Aldrich, A5040) and embedded in 1.8% low melting point agarose (Invitrogen) and subjected to imaging.

### LysoTracker staining
Two sets of 15 zebrafish larvae at 2, 3 and 5 dpf were transferred to a round-bottom 2 ml Eppendorf tube. LysoTracker™ Red DND-99 (1 mM; Invitrogen, L7528) was diluted 1:100 with E3 containing 0.003% (m/v) PTU. Subsequently, E3 medium was removed from the round-bottom 2 ml Eppendorf tube and zebrafish were incubated in 250 µl of the 10 µM Lysotracker solution at 28°C for 40 min in the dark with the caps opened. Next, zebrafish were washed and incubated in E3 containing 0.003% (m/v) PTU for 20 min at 28°C in the dark, before embedding and imaging.

### Drug treatments
Zebrafish eggs were collected shortly after the first eggs were laid, within a 15-min time window. Details on drugs, start of treatment, concentrations and drug refreshments are listed in Table S3. During drug treatment, 50 zebrafish embryo/larvae were kept in a total volume of 25 ml to assure normal embryonic development. At 3 dpf, drugs were removed and zebrafish were subjected to imaging.

### Image acquisition
For body length and brightfield images of the lateral view of the head, zebrafish were first anesthetized in 0.016% tricaine and transferred to 10% methylcellulose (Sigma-Aldrich, M7027), allowing correct positioning of the zebrafish during imaging. Images of the whole zebrafish and lateral view of the head were acquired with a Leica M165 FC microscope using a 10× dry objective and a Leica DFC550 camera. For Neutral Red images, the same microscope was used and serial images (two to four images per zebrafish brain) in the z-plane were acquired.

*In vivo* confocal imaging of zebrafish was performed using a Leica SP5 intravital microscope equipped with a 20×/1.0 NA water dipping objective using 488 (GFP), 514 (mVenus) and 561 (DsRED, LysoTracker) lasers. z-stacks with z-step size ranging from 1 to 4 µm were acquired.

## Time-lapse imaging

For overnight time-lapse imaging of the excitatory (*vglut2:*DsRED) and inhibitory (*gad1b:*GFP) neurons and Ca$^{2+}$ imaging (*elavl3:*NLS-GCaMP6s), 55 hpf zebrafish embryo were anesthetized with 0.5 mg/ml α-bungarotoxin (Invitrogen, B1601). This toxin binds irreversibly to the neuromuscular junctions (nicotinic acetylcholine receptors), thereby retaining normal (brain) development of the zebrafish embryo. Zebrafish were embedded in 0.8% low melting point agarose in a InViSPIM lattice pro sample holder. Overnight imaging was performed with the InViSPIM lattice pro (Bruker) at 28°C. For both excitatory and inhibitory neurons, every 10 min the entire brain of each zebrafish was imaged through the *z*-plane from 63 hpf to 75 hpf.

## Image and statistical analysis

Before genotyping, all images were processed and analyzed with Fiji ImageJ software. Total body lengths were measured from jaw to fin tail in zebrafish imaged from the dorsal side. Apoptotic clusters (*ubb:*SecA5-mVenus$^+$ clusters; secreted Annexin A5 fused to YFP) and oligodendrocyte precursor cells (OPCs; *olig1:*NLS-mApple) were analyzed with the 3D object counter plugin. Thresholds were same for each image of each experiment and dpf, voxel size minimum value was 4. For excitatory (*vglut2:*DsRED) and inhibitory (*gad1b:*GFP) neurons, a threshold was set on a *z*-projection of the brain. The same threshold was used for each image of each experiment and dpf, and total midbrain area was calculated. Excitatory and inhibitory neurons numbers were manually counted in a region of interest in the middle of the spinal cord images. Radial glial cell fibers (*her4.3:*EGFP) were manually counted in a region of interest in the right hemisphere of the midbrain. Microglia (Neutral Red$^+$ or *mpeg1:*EGFP$^+$) were manually counted in the whole midbrain area. LysoTracker and Ca$^{2+}$ (*elavl3:*NLS-GCaMP6s) intensities were measured as the mean fluorescence intensity on *z*-projections of the midbrain area. To analyze LysoTracker area in microglia and microglia (*mpeg1:*EGFP$^+$) circularity, a threshold was applied on the *z*-projections of midbrain regions. Six microglia per zebrafish brain were selected for morphological analysis and LysoTracker area calculations.

## Acknowledgements

We thank Dr Thomas Simmen for the fruitful discussions on TMX2. We thank the Erasmus Optical Imaging Centre (OIC) for their support.

## Competing interests

The authors declare no competing or financial interests.

## Author contributions

Conceptualization: J.D., R.S., L.E.S., G.M.S.M., T.J.v.H.; Data curation: J.D.; Formal analysis: J.D., T.J.v.H.; Investigation: J.D., W.L., R.S., F.O., H.C.v.d.L., G.-J.K., W.M.B., C.d.K., G.M.v.W., G.M.S.M., T.J.v.H.; Methodology: J.D., W.L., F.O., H.C.v.d.L., G.-J.K., W.M.B.; Project administration: J.D., T.J.v.H.; Resources: H.C.v.d.L.; Supervision: J.D., H.C.v.d.L., L.E.S., G.-J.K., G.M.v.W., G.M.S.M., T.J.v.H.; Validation: C.d.K.; Visualization: J.D., F.O.; Writing – original draft: J.D.; Writing – review & editing: J.D., W.L., R.S., H.C.v.d.L., L.E.S., G.M.S.M., T.J.v.H.

## Funding

T.J.v.H. was funded by an Erasmus University Rotterdam (EUR) fellowship. Open Access funding provided by Erasmus University Rotterdam. Deposited in PMC for immediate release.

## Data and resource availability

All relevant data and details of resources can be found within the article and its supplementary information.

## Peer review history

The peer review history is available online at https://journals.biologists.com/dev/lookup/doi/10.1242/dev.204348.reviewer-comments.pdf

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
