## [Peer Review File · Development (Cambridge, England)]

The non-canonical thioreductase Tmx2b is essential for neuronal survival during zebrafish embryonic brain development

Jordy Dekker, Wendy Lam, Herma C. van der Linde, Floris Ophorst, Charlotte de Konink, Rachel Schot, Gert-Jan Kremers, Leslie E. Sanderson, Woutje M. Berdowski, Geeske M. van Woerden, Grazia M. S. Mancini and Tjakko J. van Ham
DOI: 10.1242/dev.204348

Editor: Francois Guillemot

Review timeline

Original submission:	23 August 2024
Editorial decision:	16 October 2024
First revision received:	24 April 2025
Editorial decision:	13 May 2025
Second revision received:	5 August 2025
Accepted:	17 August 2025

Original submission

First decision letter

MS ID#: dev.204348

MS TITLE: The non-canonical thioreductase TMX2 is essential for neuronal survival during embryonic brain development

AUTHORS: Jordy Dekker, Wendy Lam, Rachel Schot, Floris Ophorst, Herma C van der Linde, Leslie Sanderson, Gert-Jan Kremers, Woutje M Berdowski, Charlotte de Konink, Geeske M van Woerden, Grazia MS Mancini and Tjakko J van Ham

Dear Dr van Ham,

I have now received all the referees' reports on the above manuscript, and have reached a decision. The referees' comments are appended below, or you can access them online: please go to:

As you will see, the referees express great interest in your work, but they also have significant criticisms and recommend a substantial revision of your manuscript before we can consider publication. In particular, they request that you validate the crispr mutants by investigating whether their phenotypes can be restored by expression of tmx2b or human TMX; that you outcross mutants for several generations; that experimental larvae come from at least two biological replicates; that you increase N in experiments where N=1; and that you investigate whether the mutated tmx2b transcript undergoes nonsense mediated RNA decay or is translated to a truncated protein that retains its localization signal.

However some of the experiments requested by referee 1 in particular, appear beyond the scope of this study and are not required for the revision of the manuscript. This is the case of the generation of a tagged variant of Tmx2b; of the investigation of the expression levels of UPR proteins; of the analysis of mitochondrial structure by EM; and of the analysis of the expression of tmx2a and tmx2b by in situ hybridization. To address the concern of the referee regarding the expression of tmx2a and tmx2b at appropriate stages, I recommend that you mention in the manuscript that this data is publicly available through ZFIN.

If you are able to revise the manuscript along the lines suggested, which may involve further experiments, I will be happy receive a revised version of the manuscript. Your revised paper will be re-reviewed by one or more of the original referees, and acceptance of your manuscript will depend on your addressing satisfactorily the reviewers' major concerns. Please also note that Development will normally permit only one round of major revision.

If it would be helpful, you are welcome to contact us to discuss your revision in greater detail. Please send us a point-by-point response indicating your plans for addressing the referees' comments, and we will look over this and provide further guidance.

Please attend to all of the reviewers' comments and ensure that you clearly highlight all changes made in the revised manuscript. Please avoid using 'Tracked changes' in Word files as these are lost in PDF conversion. I should be grateful if you would also provide a point-by-point response detailing how you have dealt with the points raised by the reviewers in the 'Response to Reviewers' box. If you do not agree with any of their criticisms or suggestions please explain clearly why this is so.

Reviewer 1

SUMMARY OF THE ADVANCE MADE IN THIS PAPER AND ITS POTENTIAL SIGNIFICANCE TO THE FIELD

The manuscript submitted by Dekker et al. provides novel insights into the function of Tmx2b showing that this factor is essential for neuronal survival during zebrafish brain development. Using CRISPR/Cas9, the authors generated two frameshift mutations in *tmx2a* and *tmx2b*, both homologs to human TMX2. Following data mining of existing RNAseq data sets as well as in house RNAseq data of wild-type zebrafish brain at 5 day post fertilization, the authors restrict their analysis to *tmx2b* because *tmx2a* is not expressed in any neuronal tissue in zebrafish.

Subsequent *tmx2b* mutant analysis is carried out using various transgenic lines and time-lapse imaging. The authors find that *tmx2b* mutants initially develop completely normally, but fail to exhibit locomotor activity at 3 days post fertilization, which is accompanied by massive cell death of postmitotic neurons in the brain. Moreover, the authors show that calcium signaling is significantly increased, indicating that dysregulation of calcium homeostasis underlies the *tmx2b* mutant brain phenotype.

Overall, the authors present interesting and important new information for *tmx2b* in zebrafish that even might serve as a disease model mimicking the phenotype of TMX2 loss in humans. However, in my view, the attempt to establish this new mutant as a novel disease model fails and the whole manuscript requires some major revisions to be considered for publication. Below I list my points to improve the study in general and the manuscript in particular.

SUGGESTIONS TO AUTHORS

Major comments:

- Expression of *tmx2a* and *tmx2b*

The authors claim that *tmx2b* is expressed ubiquitously but only provide data from existing RNAseq data sets obtained from various regeneration experiments (Fig S3A) which for sure does not cover the time frame of the *tmx2b* mutant analysis (0-5 days post fertilization). The authors also provide data from an in house RNAseq data of wild-type zebrafish brain at 5 day post fertilization (Fig S3B) but how this data was obtained remains unclear. The strongest evidence for ubiquitous expression of *tmx2b* is the simple reference to expression studies in other species, but all these species have only one Tmx2 ortholog. Instead of using some existing RNAseq data sets obtained from various regeneration experiments, the authors should use in situ hybridization and show the expression of *tmx2a* and *tmx2b* at the appropriate period (0-3 days post fertilization). In this context, the already deposited data of *tmx2a* at ZFIN (<http://zfin.org/ZDB-GENE-050208-95/expression>) shows far more convincingly that *tmx2a* is not expressed in any neural tissue during zebrafish development than any RNAseq data set can do. The expression studies should be extended to the use of transgenic

lines labeling specific cell types (e.g. radial glia using her4.3:EGFP) to clearly show the presence or absence of tmx2b in the specific cell types.

- Localization wild-type and mutant Tmx2b

In order to establish a new disease model it would be necessary to show that the zebrafish ortholog Tmx2b is distributed within the cell in the same manner as human TMX2. To this aim, it should be easy to generate a tagged variant of Tmx2b and show the presence in the endoplasmic reticulum and/or nuclear pore in the early zebrafish embryo. Moreover, a decent characterization of the new mutant tmx2b allele is required. Right now, the authors claim that they do not know if the mutated transcript undergoes nonsense mediated RNA decay or is translated to a truncated protein that retains its localization signal (Fig S2A). This issue should be addressed.

- Expression levels of UPR proteins

The authors state that TMX2 knockdown affects the expression levels of UPR proteins in mouse cortical neurons and human cholangiocarcinoma cells in vitro. To strengthen their idea that the novel tmx2b allele can serve as a new disease model, the authors should address this.

- Proofreading of the manuscript

Overall, I have to state that it was not fun to read the submitted manuscript; it contains inaccurate descriptions, missing explanations as well as typos. It would take too much time to list all items here; hence, I address only a few below.

Example for inaccurate descriptions: The authors write 'At 2 dpf the excitatory and inhibitory neurons were similar between tmx2b^{-/-} and tmx2b^{+/?} (Figure 2A-D and Figure S7&8).' However, I assume that the authors want to say 'At 2 dpf, the number of vglut2:DsRED-positive excitatory and the number of gad1b:GFP-positive inhibitory neurons were similar between tmx2b^{-/-} and tmx2b^{+/?} (Figure 2A-D and Figure S7&8).'

Example for missing explanations: The authors write 'Next, we assessed apoptosis and visualized apoptotic cells (ubb:SecA5-mVenus⁺ clusters) in zebrafish brain at 2, 3 and 4 dpf (Figure 1H,I and Figure S6).' A brief line description would help to understand what was done. Of course, the reader can search for the ubb:SecA5-mVenus line in ZFIN and figure out the function of the line on his/her own. However, the authors should actually provide this information. In the context of the ubb:SecA5-mVenus line, there is no information on how 'cluster' is defined?

Example for typo: The authors write 'As Tmx2 deficiency is early lethal in zebrafish and embryonic lethal in mice, we tested whether Tmx2 is important for survival of mammalian neurons in vitro.' I assume the authors forgot to eliminate this sentence from the manuscript because the announced experiments are not contained in the study. There are multiple other examples of missing words and punctuations.

Minor comments:

- Phenotypic analysis of tmx2a mutant allele

The authors describe the production of a tmx2a mutant allele but nothing more. Is there a phenotype? If yes, also in the brain? What about tmx2a; tmx2b double mutants? Do they show a stronger brain phenotype than tmx2b single mutants? Here, also negative data would strengthen the statements of the authors to restrict themselves to the tmx2b mutant analysis.

- Survival of tmx2b mutants

Please clarify if tmx2b mutants survive for 5-10 days post fertilization as stated in the results or do not survive beyond 5 days post fertilization as mentioned in the discussion.

- Use of abbreviations

The authors need to double check the use of abbreviations and should stick to general usage of abbreviations in scientific writing. In some cases, abbreviations are introduced but only rarely used (e.g. MCD, NDD, KO, MERC). In other cases, abbreviations are used but never introduced (e.g. WT, LoF).

- Use of the term "newborn neurons"

The authors should familiarize themselves with the term "newborn neurons" in the literature. Neurons born at 1 day post fertilization are not newborn any longer on 3 days post fertilization. I assume that the authors always mean "postmitotic neurons" but it's not always clear.

- Nomenclature

The authors should familiarize themselves with zebrafish and human nomenclature. Right now it appears random. For instance, the title contains the human TMX2 although it should contain the zebrafish variant Tmx2b because that is what the manuscript is about. Other examples in the text: calnexin, importin- β .

- Missing line numbering and page numbers.

It would be better to include these in the submitted manuscript, to address issues more specifically.

Reviewer 2

SUMMARY OF THE ADVANCE MADE IN THIS PAPER AND ITS POTENTIAL SIGNIFICANCE TO THE FIELD

Decker et al. use CRISPR/Cas9 to introduce mutations in the two zebrafish orthologs of TMX2, a thioredoxin-related transmembrane related protein. Genetic variants in human TMX2 cause MCD, microcephaly, and polymicrogyria. The goal of this paper was to begin to characterize the cellular phenotypes and understand mechanistically how mutation can result in neurodevelopmental defects. The authors created mutations in both paralogs but eloquently provide rationale for the focus on only the tmx2b gene in zebrafish. They discover that excitatory and inhibitory neurons decrease at 3 days post fertilization after being relatively normal at 2 days. They show that these populations of neurons are undergoing cell death. They also analyze radial glial cells, OPCs, and myelination. They find that radial glial cells are not affected somewhat contradictory to what has been hypothesized and is presented in the discussion. Myelination is reduced and the number of OPCs is increased. Microglial cells are increased suggesting this is a result of cell death. They further seek a mechanism, analyzing ROS and calcium influx due to the known function of the gene. They find Calcium signaling to be increased at the onset of the cell death. These data are the first to describe the cellular and molecular changes, due to the fact that mice are embryonic lethal. It is comprehensive analysis and the overall results are important to the fields of genetics, human genetics, and development. Functional analysis is a prerequisite towards diagnosis and treatment of birth defects.

SUGGESTIONS TO AUTHORS

1. It might be helpful to add some information/background about Ca signaling that is known to impair cortical development in the introduction.
2. The discussion could benefit by describing if other TMX family members are implicated in brain disease/development.
3. The description of mouse models and limitations needs to be provided in the introduction as rationale for the zebrafish system.
4. There is a line in the intro that is incorrect "as glia cells are unaffected" That is not supported. I think you mean radial glia, which are more generally stem cells in fish than otherwise. This should be corrected. There are better astrocyte markers to bolster this claim if required. The RGCs are not affected but myelin is inferring defects in glia.
5. The phenotypes must be restored by expression of tmx2b or human TMX for the crispr to be valid.
6. It is unclear what generation of the mutant that experiments were started. It is important to outcross several generations to reduce off-target effects.
7. Color of Fig. 1G too difficult to differentiate. Consider revising.
8. the idea of non-progressive does not seem supportive. Calcium increases are indeed visually progressive. and if the cause it would appear it is progressive. I am not sure the data fully support this claim.
9. what is the expression of tmx2b in the spinal cord
10. Fig. 2 begins the concerning trend of N=1 used to obtain sibling and mutant. Does this mean a single clutch with single parents. Given the genetic heterogeneity of the fish and the use of CRISPR

all experimental larvae need to come from at least two clutches on 2 different occasions. This will take substantial effort but will increase rigor.

11. Referring to RGCs as glia may be misleading and a misnomer. Conclusion under "Glia cell populations...TMX2 loss" should be revised
12. Can you clarify the age of the embryo/larvae from timelapse that the phenotype is observed. Statistical analysis of Fig 4B is required.
13. There is a lack of controls on the antioxidant experiments. These need to be repeated with controls for uptake etc. The discussion is insufficient to completely rule out this mechanism.
14. Fig5 has concerns for N=1, a typo B to E. Are the numbers in A the same as those used in B and C. Please consider different colors in C
15. There is no discussion between the contradictory phenotypes of Ca in D2 and D3, decrease and then increase, please describe
16. Data does not support the sentence "cell death occurred specifically in neurons, while OPC we decreased in number without defect. No cell death or proliferation assays are performed to support the OPC/proliferation defects
17. Have you considered analyze mitochondrial structure through EM?
18. There is a contradiction in conclusions/data between S10 and figure 3C. Seems that OPC are reduced at 2 Days. More mildly yes, but definitely reduced. Again must address the N=1 in supplementary figure

Reviewer 3

This work establishes a zebrafish model of biallelic TMX2 disruption, which causes a severe neurodevelopmental syndrome in humans. As knockout mice are embryonically lethal, this study is thus the first time the impact of this gene has been studied in a developing brain. The authors extensively characterize the impact of this mutation on excitatory neurons, inhibitory neurons, OPCs, glia, and spinal cord neurons, discovering a CNS-specific loss of newborn neurons. They narrowed down the time window of neuronal death, finding that it occurs suddenly at 3 dpf and rapidly over only 1.5 hours. Strikingly, they found that neuronal activity drove this cell death. This is a thorough and detailed manuscript, with high-quality data and robust analysis. It is also a very clearly and carefully written manuscript. The level and content of the introductory background was appropriate. The inclusion of details such as the time of day experiments were performed indicates exceptional rigor. It could have been proofread a little more carefully, however, as there were some minor formatting errors. Taken together, I am convinced of the significance of the study, that the data mostly supports the conclusions, and that the suggested mechanisms in the discussion are plausible. Several limitations of the study are already clearly laid out in the discussion.

The image in Fig 1E indicate that they also have heart edema at 3 dpf. The manuscript could benefit from an additional summary of non-neuronal phenotypes other than body length. I wonder to what extent the neuronal phenotypes are affected by decline of other organ systems? Do the authors think that the tricaine experiment indicates the brain necrosis is the primary issue and all of the changes to the rest of the body are secondary to the brain phenotype?

Is there another way to suppress neuronal excitation other than tricaine to nail down whether seizure suppression or an "additional function of voltage gated sodium channels in microglia" or some other known function could be causing the rescue? It is interesting that earlier seizure induction didn't accelerate the phenotype. Is there another way of inducing calcium dysregulation that could possibly accelerate the phenotype? What about causing seizures in another way such as with PTZ? If there are no straightforward, good experiments that would help tease apart how this rescue is happening, then I do not think it is required for publication. The phenotype and rescue are exciting, however, and if there are a couple of additional tests that could be revealing, then I think it would be worth adding.

Minor comments

- "The last 30 years many genetic causes for MCD". Missing a word.
- Use prime symbols on the primer sequences listed in the methods (CRISPR-Cas9 genome editing section).
- Use the micro symbol on microliter, not a u (entire methods section).
- "with 0,5 mg/ml" should be "."

- There's what looks like a formatting error that occurred where there's a lot of "keyword" etc (CRISPR-Cas9 genome editing section). Possibly from the reference management software.
- Figure 3 legend has a couple of *tmx2b*'s that are not italicized.
- "Hence, cortical development appears entirely normal". Obviously it is not cortical development in zebrafish.
- "disease onset in *tmx2b*^{-/-} appeared sudden between". I am not sure that I would call the zebrafish mutant phenotype a "disease", but also this sentence should use "suddenly" rather than "sudden".

First revision

Author response to reviewers' comments

Dear editor,

We thank you and the reviewers for the thorough review and constructive feedback. Please find below our response including first a direct response to the suggested main point by the editor followed by a point-by-point response to the reviewers' comments.

Editor: "As you will see, the referees express great interest in your work, but they also have significant criticisms and recommend a substantial revision of your manuscript before we can consider publication. In particular, they request that you validate the crispr mutants by investigating whether their phenotypes can be restored by expression of *tmx2b* or human TMX"

Reply: *To address the latter, we performed rescue experiments using injections of *tmx2b* mRNA. Briefly, heterozygous *tmx2b* mutants were incrossed and fertilized oocytes injected with zebrafish *tmx2b* mRNA, and subsequently, the brain phenotype (easily distinguished by light microscopy) was assessed in uninjected and injected embryos, after which individual embryos were genotyped for the *tmx2b* mutation (wt, heterozygous or homozygous). The phenotype in *tmx2b* homozygous mutants is 100% penetrant: all homozygous embryos have a clearly distinguishable cloudy brain phenotype at 3 dpf (consisting mostly of AnnexinA5+ cells) as indicated in Figure 1E. In injected *tmx2b*^{-/-} embryos we observed larvae with a normal brain in 1 out of 11 *tmx2b*^{-/-} (8% for 25 ng injected), 2 out of 6 *tmx2b*^{-/-} embryos (33% for 50 ng injected) and 3 out of 6 *tmx2b*^{-/-} (50% for 100 ng injected). Therefore, there appears to be a dose-dependent rescue of the cloudy brain phenotype. This further supports the fact that *tmx2b* loss of function is responsible for the acute brain phenotype at 3 dpf. We have added these data in Suppl figure S5 and added explanation to the results section (page 10, lines 236-246). We have also tried to independently target the *tmx2b* locus by CRISPR/Cas9, but unfortunately directly targeting the locus was not sufficiently efficient to induce the phenotype.*

*Of note, we noticed that overexpression of *tmx2b* caused some toxicity as apparent from malformed embryos that was independent of genotype, but despite the toxicity rescue of the brain phenotype could be observed.*

Editor: "that you outcross mutants for several generations;"

Reply: *Mutants were outcrossed twice before the first experiments were carried out (Fig 1), and additional experiments were carried out with lines outcrossed at least 3 times. Furthermore, all experiments were carried out in larvae derived from heterozygous incrosses, with sibling controls, which makes it unlikely that another genetic factor could segregate with the phenotype. Also, the *tmx2* cloudy brain phenotype can be recognized easily through light microscopy, and this was seen in all experiments and confirmed by genotyping. Together, it is highly unlikely that an off-target mutation on one of the chromosomes could cause the phenotype as segregation of phenotype and genotype in all the experiments performed excludes this possibility. The rescue experiments also exclude the possibility that an off-target mutation nearby *tmx2b* on the same chromosome would cause the phenotype.*

Editor: "that experimental larvae come from at least two biological replicates; that you increase N in experiments where N=1;"

Reply: We have indicated in several instances that N=1 where only one quantitative experiment was carried out. Below we provide our argumentation explaining why in our opinion this is a valid approach here. The experiments were often carried out quantifying the phenotype on multiple consecutive days (1) in groups of larvae from different clutches/different parents (2) from heterozygous incrosses (3). In our experience in particular (3), using heterozygous incrosses, yields very robust data by minimizing differences in genetic background. Furthermore differences in cellular phenotypes are quite large, imaging almost the entire brain

(4) and most of the experiments investigate the brain phenotype from different angles (5): The “cloudy brain” phenotype is 100% penetrant, always occurring at 72 hour post fertilization. Consistent with cell death in the brain at 72 hpf we see a massive increase in AnnexinA5+ apoptotic cells at 72 hpf. Next, having shown by 2 independent techniques that brain cells appear to die in large numbers, we look at two widely expressed neuronal markers at 2 (N=1), 3 (N=2) and 5 dpf (N=1) as this is, next to radial glia, the most prevalent cell type at that stage. As could be expected (based on the patient phenotype and on the massive cell death appearing only at 3 dpf) we see no difference in brain area—visual pattern and quantitatively—at 2 dpf using two different neuronal markers—for excitatory as well as inhibitory neurons—, a difference at 3 dpf were mutants have decreased neuronally marked area, and an even larger difference at 5 dpf.

Having shown two types of neurons are affected in *tmx2b*^{-/-} at the precise stage where we see cell death, we investigated radial glia another prevalent cell type. Surprisingly, we do not see differences in radial glia—visual pattern looks the same, and quantification shows no difference—, at three different time points 2, 3 and 5 dpf, investigating this with in vivo imaging in at least 11 animals. We performed this experiment only once.

We next investigated two more cell types that are less frequent in the brain, more differentiated cells and these show a large decrease in numbers (*olig1*⁺) or myelinated area (*mbp*:GFP-CAAX⁺). For *mbp*:EGFP-CAAX, labeling myelinated axons this is to be expected: if large numbers of brain cells die, the sensitive oligo’s relying on neuronal connections for their establishment would also be affected. For *olig1*⁺ cells there is no difference at 2 dpf (consistent with all other measurements that the brain is normal at this stage) and a large difference at 3 dpf (*olig1*⁺ cells increase in controls, which does not happen in mutants). Lastly, we looked at microglia, and these showed a minor increase in numbers, and have different morphology as we always observe when there is cell death occurring in the brain (e.g. Oosterhof et al., *Glia*, 2015, Sanderson et al., *Brain*, 2021).

The researchers in my group characterized these phenotypes and meticulously planned these experiments to acquire the relevant data in one large experiment. The transgenic lines we use here are well characterized in our lab and we have a good overview of the variability and extent of phenotypes (Kuil et al., 2019, *Glia*, Sofou et al., 2021, *Embo Mol Med*, Berdowski et al., 2022, *Acta neuropathologica*, Smits et al., 2023, *Hum Genet*, 2023). In all honesty, in the 6 papers from my group mentioned, assessing largely similar phenotypes in a similar manner, we did not indicate how often each experiment was carried out—often multiple times, first a small pilot and then the full experiment, sometimes perhaps once—. The current first author, who wrote the first draft, did a good job in giving full disclosure on the experimental design, however we feel that repeating experiments mentioned would not affect the conclusion of this work and does not have the potential to add additional insight, and last these are very laborious experiments. Altogether, we propose to move the “N = number of experiments” in the materials and methods and/or provide a brief explanation on the rationale behind our approach. We hope you agree this would be an acceptable change to the manuscript, together with the additional experiments included in the revised manuscript, being sufficient to support our current conclusions.

Editor: “and that you investigate whether the mutated *tmx2b* transcript undergoes nonsense mediated RNA decay or is translated to a truncated protein that retains its localization signal.”

Reply: We performed additional experiments to assess whether *tmx2b* transcript undergoes nonsense mediated mRNA decay (Figures S4G).

Editor: However some of the experiments requested by referee 1 in particular, appear beyond the scope of this study and are not required for the revision of the manuscript. This is the case of the generation of a tagged variant of *Tmx2b*; of the investigation of the expression levels of UPR proteins; of the analysis of mitochondrial structure by EM; and of the analysis of the expression of *tmx2a* and *tmx2b* by in situ hybridization. To address the concern of the referee regarding the expression of *tmx2a* and *tmx2b* at appropriate stages, I recommend that you mention in the manuscript that this data is publicly available through ZFIN.

Reply: We agree with the editor that these experiments are beyond the scope of the current manuscript. *In situ* hybridisations for *tmx2b* were not available from ZFIN, and for *tmx2a* only a specific early developmental stage, however RNAseq data of multiple sources indicates expression of *tmx2b* in relevant developmental stages (e.g. at 2, 3 and 5 dpf)

To submit your revised manuscript, please go to:

<https://www.editorialmanager.com/develop/> and click on the 'Submissions Needing Revision' within the Author Main Menu.

Reviewer 1: The manuscript submitted by Dekker et al. provides novel insights into the function of Tmx2b showing that this factor is essential for neuronal survival during zebrafish brain development. Using CRISPR/Cas9, the authors generated two frameshift mutations in *tmx2a* and *tmx2b*, both homologs to human TMX2. Following data mining of existing RNAseq data sets as well as in house RNAseq data of wild-type zebrafish brain at 5 day post fertilization, the authors restrict their analysis to *tmx2b* because *tmx2a* is not expressed in any neuronal tissue in zebrafish.

Subsequent *tmx2b* mutant analysis is carried out using various transgenic lines and time-lapse imaging. The authors find that *tmx2b* mutants initially develop completely normally, but fail to exhibit locomotor activity at 3 days post fertilization, which is accompanied by massive cell death of postmitotic neurons in the brain. Moreover, the authors show that calcium signaling is significantly increased, indicating that dysregulation of calcium homeostasis underlies the *tmx2b* mutant brain phenotype. Overall, the authors present interesting and important new information for *tmx2b* in zebrafish that even might serve as a disease model mimicking the phenotype of TMX2 loss in humans. However, in my view, the attempt to establish this new mutant as a novel disease model fails and the whole manuscript requires some major revisions to be considered for publication. Below I list my points to improve the study in general and the manuscript in particular

SUGGESTIONS TO AUTHORS

Major comments:

- Expression of *tmx2a* and *tmx2b*

The authors claim that *tmx2b* is expressed ubiquitously but only provide data from existing RNAseq data sets obtained from various regeneration experiments (Fig S3A) which for sure does not cover the time frame of the *tmx2b* mutant analysis (0-5 days post fertilization). The authors also provide data from an in house RNAseq data of wild-type zebrafish brain at 5 day post fertilization (Fig S3B) but how this data was obtained remains unclear. The strongest evidence for ubiquitous expression of *tmx2b* is the simple reference to expression studies in other species, but all these species have only one Tmx2 ortholog. Instead of using some existing RNAseq data sets obtained from various regeneration experiments, the authors should use *in situ* hybridization and show the expression of *tmx2a* and *tmx2b* at the appropriate period (0-3 days post fertilization). In this context, the already deposited data of *tmx2a* at ZFIN (<http://zfin.org/ZDB-GENE-050208-95/expression>) shows far more convincingly that *tmx2a* is not expressed in any neural tissue during zebrafish development than any RNAseq data set can do. The expression studies should be extended to the use of transgenic lines labeling specific cell types (e.g. radial glia using her4.3:EGFP) to clearly show the presence or absence of *tmx2b* in the specific cell types.

Reply: We agree that *tmx2a* *in situ* hybridization shows no expression in the nervous system, although only up to ~1,5 dpf. We have added this information to the manuscript. According to the reviewer the zfRegeneration: "which for sure does not cover the time frame of the *tmx2b* mutant analysis (0-5 days post fertilization)." This is incorrect, as at least one of the included studies is the "Zebrafish transcriptome sequencing project" carried out by the Wellcome Trust Sanger Institute, showing very low expression of *tmx2a* at 2, 3 and 5 dpf, and much higher expression of *tmx2b*.

- Localization wild-type and mutant Tmx2b.

In order to establish a new disease model it would be necessary to show that the zebrafish ortholog Tmx2b is distributed within the cell in the same manner as human TMX2. To this aim, it should be easy to generate a tagged variant of Tmx2b and show the presence in the endoplasmic reticulum and/or nuclear pore in the early zebrafish embryo. Moreover, a decent characterization of the new mutant *tmx2b* allele is required. Right now, the authors claim that they do not know if the mutated

transcript undergoes nonsense mediated RNA decay or is translated to a truncated protein that retains its localization signal (Fig S2A). This issue should be addressed.

Reply: *we have tested whether the mutant transcript undergoes nonsense mediated mRNA decay. This showed that the transcript with the mutation was almost not detectable. We have included this as supplemental Figure S4, panel G.*

- Expression levels of UPR proteins

The authors state that TMX2 knockdown affects the expression levels of UPR proteins in mouse cortical neurons and human cholangiocarcinoma cells in vitro. To strengthen their idea that the novel tmx2b allele can serve as a new disease model, the authors should address this.-

Reply: *We thank the reviewer for this suggestion, but we believe it is beyond the scope of this manuscript to confirm these previous results on UPR proteins in this current manuscript.*

Proofreading of the manuscript.

Overall, I have to state that it was not fun to read the submitted manuscript; it contains inaccurate descriptions, missing explanations as well as typos. It would take too much time to list all items here; hence, I address only a few below. Example for inaccurate descriptions: The authors write 'At 2 dpf the excitatory and inhibitory neurons were similar between tmx2b^{-/-} and tmx2b^{+/?} (Figure 2A-D and Figure S7&8).' However, I assume that the authors want to say 'At 2 dpf, the number of vglut2:DsRED-positive excitatory and the number of gad1b:GFP-positive inhibitory neurons were similar between tmx2b^{-/-} and tmx2b^{+/?} (Figure 2A-D and Figure S7&8).'

Reply: *We are not sure what the reviewer intends to say with that last part. Arguably, it could be considered sloppy use of language to refer to cells positive for vglut2-DsRED as excitatory neurons, as they simply express a single marker which does not always overlap with the identity of the cells. The suggestion to use "vglut2:DsRED positive excitatory neurons" does not have our preference as this suggests that we are looking at a subpopulation of excitatory neurons that are vglut2+, and that is possible but we cannot be certain of that.*

We agree with the reviewer that it is better to use the transgenic line to indicate the marker than to assume the cells are excitatory or inhibitory. We have changed this throughout the manuscript.

Example for missing explanations: The authors write 'Next, we assessed apoptosis and visualized apoptotic cells (ubb:SecA5-mVenus⁺ clusters) in zebrafish brain at 2, 3 and 4 dpf (Figure 1H,I and Figure S6).' A brief line description would help to understand what was done. Of course, the reader can search for the ubb:SecA5-mVenus line in ZFIN and figure out the function of the line on his/her own. However, the authors should actually provide this information. In the context of the ubb:SecA5-mVenus line, there is no information on how 'cluster' is defined?

Reply: *we thank the reviewer for spotting this. The last author has used this transgenic line for 15 years, and overlooked this issue. We have added a brief explanation of Annexin A5 as an apoptosis marker at page 8 line 189.*

Example for typo: The authors write 'As Tmx2 deficiency is early lethal in zebrafish and embryonic lethal in mice, we tested whether Tmx2 is important for survival of mammalian neurons in vitro.' I assume the authors forgot to eliminate this sentence from the manuscript because the announced experiments are not contained in the study. There are multiple other examples of missing words and punctuations.

Reply: *we apologize for not having seen this before. We have corrected and removed these.*

Minor comments:

Phenotypic analysis of tmx2a mutant allele. The authors describe the production of a tmx2a mutant allele but nothing more. Is there a phenotype? If yes, also in the brain? What about tmx2a; tmx2b double mutants? Do they show a stronger brain phenotype than tmx2b single mutants? Here, also negative data would strengthen the statements of the authors to restrict themselves to the tmx2b mutant analysis.

Reply: *tmx2a mutant did not show a phenotype as explained at page 9 line 209. We have not tested if the tmx2a zebrafish exhibit a stronger phenotype.*

Survival of tmx2b mutants

Please clarify if tmx2b mutants survive for 5-10 days post fertilization as stated in the results or do not survive beyond 5 days post fertilization as mentioned in the discussion.

Reply: *The $tmx2b^{-/-}$ eventually die because they are unable feed themselves, since they are unable to move/swim. So, our $tmx2b^{-/-}$ zebrafish die after reaching the feeding stage. We have changed this in the discussion at page 17 line 414-419: “Most affected individuals present either with biallelic missense variants or with compound heterozygous truncating, i.e. null mutation, and a missense variant. $Tmx2$ knockout mice are embryonically lethal and our $tmx2b^{-/-}$ zebrafish also do not survive after reaching the feeding stage”*

Use of abbreviations

The authors need to double check the use of abbreviations and should stick to general usage of abbreviations in scientific writing. In some cases, abbreviations are introduced but only rarely used (e.g. MCD, NDD, KO, MERC). In other cases, abbreviations are used but never introduced (e.g. WT, LoF).

Reply: *We agree with the reviewer and have modified this throughout the manuscript.*

Use of the term "newborn neurons" The authors should familiarize themselves with the term "newborn neurons" in the literature. Neurons born at 1 day post fertilization are not newborn any longer on 3 days post fertilization. I assume that the authors always mean "postmitotic neurons" but it's not always clear.

Reply: *We have exchanged newborn neurons for post-mitotic neurons.*

Nomenclature

The authors should familiarize themselves with zebrafish and human nomenclature. Right now it appears random. For instance, the title contains the human TMX2 although it should contain the zebrafish variant $Tmx2b$ because that is what the manuscript is about. Other examples in the text: calnexin, importin- β .

Reply: *Regarding the title, we are interested in using zebrafish to help understand the role of human TMX2. We agree with the reviewer that we investigated $Tmx2b$ and therefore more appropriate to include this in the title. Therefore, we have changed the title to: “The non-canonical thioreductase $Tmx2b$ is essential for neuronal survival during embryonic brain development in zebrafish”.*

Regarding calnexin and importin- β : the studies we refer to in the manuscript performed their experiments on human cells. To avoid confusion, we have added the HUGO Gene Nomenclature Committee (<https://www.genenames.org/>) approved gene/protein symbols for these human proteins throughout the introduction.

Missing line numbering and page numbers.

It would be better to include these in the submitted manuscript, to address issues more specifically.

Reply: *We now added page and line numbers.*

Reviewer 2: Decker et al. use CRISPR/Cas9 to introduce mutations in the two zebrafish orthologs of TMX2, a thioredoxin-related transmembrane related protein. Genetic variants in human TMX2 cause MCD, microcephaly, and polymicrogyria. The goal of this paper was to begin to characterize the cellular phenotypes and understand mechanistically how mutation can result in neurodevelopmental defects. The authors created mutations in both paralogs but eloquently provide rationale for the focus on only the $tmx2b$ gene in zebrafish. They discover that excitatory and inhibitory neurons decrease at 3 days post fertilization after being relatively normal at 2 days. They show that these populations of neurons are undergoing cell death. They also analyze radial glial cells, OPCs, and myelination. They find that radial glial cells are not affected somewhat contradictory to what has been hypothesized and is presented in the discussion. Myelination is reduced and the number of OPCs is increased. Microglial cells are increased suggesting this is a result of cell death. They further seek a mechanism, analyzing ROS and calcium influx due to the known function of the gene. They find Calcium signaling to be increased at the onset of the cell death. These data are the first to describe the cellular and molecular changes, due to the fact that mice are embryonic lethal. It is comprehensive analysis and the overall results are important to the fields of genetics, human genetics, and development. Functional analysis is a prerequisite towards diagnosis and treatment of birth defects.

SUGGESTIONS TO AUTHORS

1. It might be helpful to add some information/background about Ca signaling that is known to impair cortical development in the introduction.

Reply: *We added this pathway as a modifier of cortical development to the introduction (first paragraph) and quoted the following review on this topic McKinney et al., 2022, Development.*

2. The discussion could benefit by describing if other TMX family members are implicated in brain disease/development.

Reply: *Of the other TMX genes, only TMX5 genetic variants have been described to cause Meckel Gruber syndrome with a very different disease phenotype, involving cilia abnormalities (Shaheen et al., 2016, Genome Biol).. We added this information on page 3, line 57-58.*

3. The description of mouse models and limitations needs to be provided in the introduction as rationale for the zebrafish system.

Reply: *We disagree that mouse models and limitations would need to be introduced. Zebrafish are widely used in studies of the vertebrate brain/nervous system. Like mice, other model organisms including C. elegans and D. melanogaster, each have their particular advantages and it would go too far to go into this.*

4. There is a line in the intro that is incorrect "as glia cells are unaffected" That is not supported. I think you mean radial glia, which are more generally stem cells in fish than otherwise. This should be corrected. There are better astrocyte markers to bolster this claim if required. The RGCs are not affected but myelin is inferring defects in glia.

Reply: *we agree, indeed glial cells are affected, we have corrected this as follows at page 2 line 29: "Strikingly, cell death in tmx2b mutants occurs specifically in post-mitotic neurons within a ~1.5-hour timeframe, whereas neuronal progenitor and radials glial cells are preserved, and could be suppressed by inhibiting neuronal activity"*

5. The phenotypes must be restored by expression of tmx2b or human TMX for the crispr to be valid.

Reply: *see our response to the editor. We have performed additional experiments to investigate this.*

6. It is unclear what generation of the mutant that experiments were started. It is important to outcross several generations to reduce off-target effects.

Reply: *see response to the editor. We have added additional explanation regarding potential off target effects.*

7. Color of Fig. 1G too difficult to differentiate. Consider revising.

Reply: *we have adjusted the colors.*

8. the idea of non-progressive does not seem supportive. Calcium increases are indeed visually progressive. and if the cause it would appear it is progressive. I am not sure the data fully support this claim.

Reply: *We don't understand what the reviewer intends to say here.*

9. what is the expression of tmx2b in the spinal cord

Reply: *According to the Daniocell online single-cell RNA-seq database, tmx2b in the spinal cord is similar to that in neurons.*

Source: <https://daniocell.nichd.nih.gov/gene/T/tmx2b/tmx2b.html>

10. Fig. 2 begins the concerning trend of N=1 used to obtain sibling and mutant. Does this mean a single clutch with single parents. Given the genetic heterogeneity of the fish and the use of CRISPR all experimental larvae need to come from at least two clutches on 2 different occasions. This will take substantial effort but will increase rigor.

Reply: *Also see response to the editor. In experiments multiple clutches from independent crosses were used. Experiments were performed using in-crosses of multiple male and female heterozygous zebrafish in multiple breeding tanks, comparing homozygous mutants directly to heterozygous or wildtype siblings.*

11. Referring to RGCs as glia may be misleading and a misnomer. Conclusion under "Glia cell populations...TMX2 loss" should be revised

Reply: *We have adjusted the text in both the abstract and the last paragraph of the introduction and now only say that radial glia cell populations are not affected by Tmx2b loss (page 2, line 29; page 4, line 90)*

12. Can you clarify the age of the embryo/larvae from timelapse that the phenotype is observed. Statistical analysis of Fig 4B is required.

Reply: *The timelapse experiments were performed from 58 hpf to 75 hpf with image acquisition of the total brain every 10 minutes. The data of the three $tmx2b^{+/?}$ and $tmx2b^{-/-}$ zebrafish shown in figure 4 are from three independent experiments. The exact onset of neuronal cell death in hpf can differ slightly between experiments, due to natural variation. Therefore, we have synchronized the results at the first timepoint before neuronal cell death occurred in the $tmx2b^{-/-}$ zebrafish. To avoid confusion we have changed this timepoint to "0" in Figure 4. Furthermore, we have added statistical analysis to Figure 4B and a description to the legend.*

13. There is a lack of controls on the antioxidant experiments. These need to be repeated with controls for uptake etc. The discussion is insufficient to completely rule out this mechanism.

Reply: *We agree that we cannot rule out that potentially the phenotype can be explained by ROS and could be suppressed by antioxidants and we have more explicitly added this to the discussion. We chose for example for EN460 a dose previously published to show a mild effect on development indicating at least a dosage with a biological effect (1 μ M, Geldenhuys et al., 2017). Higher doses are toxic. For NAC we used 5 mM and for GKT137831 10 μ M.*

We had at that point the data showing the surprising and quite strong suppressive effect of tricaine on the phenotype with a quite low dose, and therefore did not further explore the hypothesis that ROS play a role.

14. Fig5 has concerns for N=1, a typo B to E. Are the numbers in A the same as those used in B and C. Please consider different colors in C

Reply: *The typo has been corrected. Colors of panel C have been adjusted.*

15. There is no discussion between the contradictory phenotypes of Ca in D2 and D3, decrease and then increase, please describe

Reply: *The Ca^{2+} phenotype is already mentioned in the discussion. We do not consider the Ca^{2+} phenotypes observed at 2 and 3 dpf contradictory. We suggest in the discussion that Ca^{2+} is misregulated (dyshomeostasis) in neurons and possibly also in glial cell types, which could explain the phenotype.*

16. Data does not support the sentence "cell death occurred specifically in neurons, while OPC we decreased in number without defect. No cell death or proliferation assays are performed to support the OPC/proliferation defects.

Reply: *We suggest that the OPC decrease is secondary to the loss of neurons/axons since the OPC proliferation is dependent on neuronal activity and the presence of axons as explained in the discussion at page 19 line 466-473.*

17. Have you considered analyze mitochondrial structure through EM?

Reply: *We appreciate the reviewer's suggestion regarding the analysis of mitochondrial structure through electron microscopy (EM). This and other subcellular aspects could indeed be studied by EM. However, performing high quality EM is not trivial, laborious and we currently do not have access directly to EM equipment and personnel.*

18. There is a contradiction in conclusions/data between S10 and figure 3C. Seems that OPC are reduced at 2 Days. More mildly yes, but definitely reduced. Again must address the N=1 in supplementary figure

Reply: *We are not sure what the reviewer is suggesting. The images of the zebrafish shown in each figure are those that are close to the average in the tested population. So the $tmx2b^{+/?}$ and $tmx2b^{-/-}$ zebrafish shown in Figure S10 have near similar olig1:NLS-mApple cell counts. Possibly the number of olig1:NLS-mApple cells between the two images appears different since they have a different distribution or the orientation of zebrafish during imaging is slightly different.*

Reviewer 3: This work establishes a zebrafish model of biallelic *TMX2* disruption, which causes a severe neurodevelopmental syndrome in humans. As knockout mice are embryonically lethal, this study is thus the first time the impact of this gene has been studied in a developing brain. The authors extensively characterize the impact of this mutation on excitatory neurons, inhibitory neurons, OPCs, glia, and spinal cord neurons, discovering a CNS-specific loss of newborn neurons. They narrowed down the time window of neuronal death, finding that it occurs suddenly at 3 dpf and rapidly over only 1.5 hours. Strikingly, they found that neuronal activity drove this cell death. This is a thorough and detailed manuscript, with high-quality data and robust analysis. It is also a very clearly and carefully written manuscript. The level and content of the introductory background was appropriate. The inclusion of details such as the time-of-day experiments were performed indicates exceptional rigor. It could have been proofread a little more carefully, however, as there were some minor formatting errors. Taken together, I am convinced of the significance of the study, that the data mostly supports the conclusions, and that the suggested mechanisms in the discussion are plausible. Several limitations of the study are already clearly laid out in the discussion.

The image in Fig 1E indicate that they also have heart edema at 3 dpf. The manuscript could benefit from an additional summary of non-neuronal phenotypes other than body length.

Reply: *The phenotype is remarkable in our opinion, as at 2 dpf we cannot distinguish any changes in mutants compared to wildtype zebrafish larvae. However, the severe brain phenotype at 3 dpf is accompanied by a decrease in length/in ability to grow. There appear no other abnormalities at this point.*

I wonder to what extent the neuronal phenotypes are affected by decline of other organ systems? Do the authors think that the tricaine experiment indicates the brain necrosis is the primary issue and all of the changes to the rest of the body are secondary to the brain phenotype?

Reply: *Patients with *TMX2* mutations reported have a severe brain phenotype, whereas other organs appear unaffected. Likely *TMX2* (*Tmx2b*) is also important to other tissues but possibly due to the enormous changes in cell numbers, types and developmental migration occurring in the brain, this may perhaps be most sensitive and where the phenotype becomes clear the first. Furthermore, organs including the kidney have an overcapacity before symptoms occur. Typically clinical manifestations of kidney failure are observed when its function is <15-20%.*

Is there another way to suppress neuronal excitation other than tricaine to nail down whether seizure suppression or an "additional function of voltage gated sodium channels in microglia" or some other known function could be causing the rescue? It is interesting that earlier seizure induction didn't accelerate the phenotype. Is there another way of inducing calcium dysregulation that could possibly accelerate the phenotype? What about causing seizures in another way such as with PTZ? If there are no straightforward, good experiments that would help tease apart how this rescue is happening, then I do not think it is required for publication. The phenotype and rescue are exciting, however, and if there are a couple of additional tests that could be revealing, then I think it would be worth adding.

Reply: *We thank the reviewer for this suggestion. At this moment it is unfortunately not possible for us to perform such additional experiments.*

Minor comments

- "The last 30 years many genetic causes for MCD". Missing a word.

Reply: *We have added the word "in" at the start of the sentence.*

- Use prime symbols on the primer sequences listed in the methods (CRISPR-Cas9 genome editing section).

Reply: *We thank the reviewer for spotting our misuse of apostrophes where we should have used prime symbols. We have changed accordingly.*

- Use the micro symbol on microliter, not a u (entire methods section).

Reply: *We have changed this throughout the methods section.*

- "with 0,5 mg/ml" should be "."

Reply: We thank the reviewer for noticing this mistake, we have corrected this.

- There's what looks like a formatting error that occurred where there's a lot of "keyword" etc (CRISPR-Cas9 genome editing section). Possibly from the reference management software.

Reply: Somehow this error from the reference management software was introduced when converting the word file to an Indesign file. This errors has been corrected.

- Figure 3 legend has a couple of *tmx2b*'s that are not italicized.

Reply: We thank the reviewing for noticing this. These *tmx2b* are now italicized.

- "Hence, cortical development appears entirely normal". Obviously it is not cortical development in zebrafish.

Reply: We have now changed "cortical" to "brain". Page 13 line 322. "Hence, brain development appears entirely normal in *tmx2b*^{-/-} zebrafish until the onset neuronal cell death, which occurs suddenly and spans an ~1.5 hours timeframe."

"disease onset in *tmx2b*^{-/-} appeared sudden between". I am not sure that I would call the zebrafish mutant phenotype a "disease", but also this sentence should use "suddenly" rather than "sudden".

Reply: we have modified the text as follows: "At 1 and 2 dpf *tmx2b*^{-/-} mutants responded normally to touch by showing increased movement, but they were non-responsive at 3 and 4 dpf, indicating that the visually distinguishable phenotype onset in *tmx2b*^{-/-} appeared suddenly between 2 and 3 dpf". (Figure 1G and Figure S5 and Videos S1-9)."

Second decision letter

MS ID#: dev.204348R1

MS TITLE: The non-canonical thioreductase *Tmx2b* is essential for neuronal survival during zebrafish embryonic brain development

AUTHORS: Jordy Dekker, Wendy Lam, Rachel Schot, Floris Ophorst, Herma C van der Linde, Leslie Sanderson, Gert-Jan Kremers, Woutje M Berdowski, Charlotte de Konink, Geeske M van Woerden, Grazia MS Mancini and Tjakko J van Ham

Dear Dr van Ham,

I have now received the reports of two of the referees who had reviewed the first version of your manuscript and I have reached a decision. The referees' comments are appended below, or you can access them online: please go to .

The overall evaluation is positive and we would like to publish a revised manuscript in *Development*, provided that the comments of the two referees can be satisfactorily addressed. You should in particular discuss further the different Ca²⁺ phenotypes at d2 and d3 and clarify the discussion on Ca²⁺ misregulation. You should provide more representative images in Fig. S10, and you should modify further your discussion to better address the points brought up by reviewer 3 in her first report. Please attend to all of the reviewers' comments in your revised manuscript and detail them in your point-by-point response. If you do not agree with any of their criticisms or suggestions explain clearly why this is so.

Reviewer 2

SUMMARY OF THE ADVANCE MADE IN THIS PAPER AND ITS POTENTIAL SIGNIFICANCE TO THE FIELD

The authors develop a new model to study the function of *TMX2* in brain development. The model survives to 5 days post fertilization while the murine model has embryonic lethality. The analysis shows dysregulation of calcium signaling and post-mitotic neuronal cell death. The model described

is of interest to the field of genetics and developmental biology and could be a useful tool for future mechanistic studies. However, it does not necessarily model the disease of interest. This reviewer finds that to be a minor point though given the utility of zebrafish to understand development. From the first revision, comments from R2 and R3 were partially addressed. R1 comments appear to be minimally addressed. The study would be more impactful if there was more mechanism to support the phenotypic analysis.

SUGGESTIONS TO AUTHORS

The authors wisely tried to restore the phenotypes in the mutant allele with overexpression/injection. However, there was little utility in this as the mRNA injection was primarily toxic yielding very few animals that were actually restored. The assay itself is not useful with the toxic effects. Some thoughtful alternatives or improvements to this assay remain needed.

While zebrafish are highly regarded for these types of studies, this reviewer strongly disagrees with reviewer 2 reply number 3. It is a minor point overall to include rationale for the model system and seems like a futile point for the authors to argue.

The calcium experiments and interpretation continue to be problematic without explanation and mechanistic inquiry. Differences in D2 and D3. Dysregulation is vague.

More representative images should be considered based on the reply to comment 18 R2 based on the reply from the authors.

Reviewer 3

The authors did not make changes the manuscript to address some of my comments. I would have preferred that they made some modifications to the discussion to discuss the points I brought up, and I still think there are other effects on the fish (the heart looks different in all of the side view images). However, I enjoyed the original paper and was less concerned with the issues brought up by the other reviewers because of the strength of the phenotype and intriguing tricaine rescue. Even without making additional changes, I think this manuscript is a reasonable fit for Development in its current state.

Second revision

Author response to reviewers' comments

Dear dr Guillemot, dear editor,

We thank you and the reviewers for the feedback.

“The overall evaluation is positive and we would like to publish a revised manuscript in Development, provided that the comments of the two referees can be satisfactorily addressed. You should in particular discuss further the different Ca²⁺ phenotypes at d2 and d3 and clarify the discussion on Ca²⁺ misregulation. You should provide more representative images in Fig. S10, and you should modify further your discussion to better address the points brought up by reviewer 3 in her first report. Please attend to all of the reviewers' comments in your revised manuscript and detail them in your point-by-point response. If you do not agree with any of their criticisms or suggestions explain clearly why this is so.”

Reviewer 2: SUMMARY OF THE ADVANCE MADE IN THIS PAPER AND ITS POTENTIAL SIGNIFICANCE TO THE FIELD

The authors develop a new model to study the function of TMX2 in brain development. The model survives to 5 days post fertilization while the murine model has embryonic lethality. The analysis shows dysregulation of calcium signaling and post-mitotic neuronal cell death. The model described is of interest to the field of genetics and developmental biology and could be a useful tool for future mechanistic studies. However, it does not necessarily model the disease of interest. This reviewer finds that to be a minor point though given the utility of zebrafish to understand development. From the first revision, comments from R2 and R3 were partially addressed. R1 comments appear to be minimally addressed. The study would be more impactful if there was more mechanism to support the phenotypic analysis.

SUGGESTIONS TO AUTHORS

The authors wisely tried to restore the phenotypes in the mutant allele with overexpression/injection. However, there was little utility in this as the mRNA injection was primarily toxic yielding very few animals that were actually restored. The assay itself is not useful with the toxic effects. Some thoughtful alternatives or improvements to this assay remain needed.

Reply: We agree with this reviewer that we observed rescue in few animals. However, considering the severity of the phenotype, toxicity of injected *tmx2b* and the fact the *tmx2b* expression is derived from a single dose of mRNA injected in the fertilized oocyte and seeing a clear rescue effect in some animals 3 days later this seems to us quite convincing that reconstituting *tmx2b* in this non-specific manner is sufficient and capable of rescuing the mutant phenotype.

While zebrafish are highly regarded for these types of studies, this reviewer strongly disagrees with reviewer 2 reply number 3. It is a minor point overall to include rationale for the model system and seems like a futile point for the authors to argue.

The calcium experiments and interpretation continue to be problematic without explanation and mechanistic inquiry. Differences in D2 and D3. Dysregulation is vague.

Reply: The images on 2 and 3 dpf of the Ca^{2+} reporter indeed show differences that in our opinion represent well what we observe under the microscope. At day 3 fluorescence is strongly increased in *tmx2b*^{-/-}, which could be related to apoptosis occurring at that time point which is known to involve loss of calcium homeostatic control leading to influx of Ca^{2+} . Our explanation of the change in calcium in the results was limited to the following sentence: “Our previous data could not identify abnormalities during the first 2 days of development in *tmx2b*^{-/-} zebrafish; however, these data suggest that Ca^{2+} dysregulation occurs prior onset of neuronal cell death.”.

We have removed the word dysregulation. And replaced by “data indicate that Ca^{2+} is reduced at a stage where we do not observe other cellular phenotypes thus preceding major loss of neurons to apoptosis. This suggests that diminished cytoplasmic Calcium precedes neuronal cell death. “

More representative images should be considered based on the reply to comment 18 R2 based on the reply from the authors.

Reply: We have updated Figure S10 and replaced the the 2 dpf images with 2 other representative zebrafish images at 2 dpf.

Reviewer 3: The authors did not make changes the manuscript to address some of my comments. I would have preferred that they made some modifications to the discussion to discuss the points I brought up, and I still think there are other effects on the fish (the heart looks different in all of the side view images). However, I enjoyed the original paper and was less concerned with the issues brought up by the other reviewers because of the strength of the phenotype and intriguing tricaine rescue. Even without making additional changes, I think this manuscript is a reasonable fit for Development in its current state.

Reply: We agree that the phenotype may not be limited to the brain, but this is by far the most obviously affected tissue. We have added the following sentence in the results section page 10 Line 236: “Besides the brain phenotype and growth delay, *tmx2b*^{-/-} zebrafish also developed cardiac

edema with no macroscopic evidence of involvement of other organs (Fig. 1E).” and in the discussion page 20 line 489 “The only abnormalities we observed outside of the brain in *tmx2b*^{-/-} zebrafish were cardiac edema and a minor decrease in longitudinal growth, which did not precede neuronal cell death and are therefore likely secondary to the brain developmental phenotype.”

Previous comments rev3: The image in Fig 1E indicate that they also have heart edema at 3 dpf. The manuscript could benefit from an additional summary of non-neuronal phenotypes other than body length.

Reply: we included some information on the non-neuronal phenotype.

I wonder to what extent the neuronal phenotypes are affected by decline of other organ systems? Do the authors think that the tricaine experiment indicates the brain necrosis is the primary issue and all of the changes to the rest of the body are secondary to the brain phenotype?

Reply: Indeed, based on the data our conclusion would be that the non-neuronal phenotype is secondary to the very severe brain phenotype and in addition it is possible that the function of *Tmx2b* is less critically important at this stage for other cell types.

Is there another way to suppress neuronal excitation other than tricaine to nail down whether seizure suppression or an "additional function of voltage gated sodium channels in microglia" or some other known function could be causing the rescue? It is interesting that earlier seizure induction didn't accelerate the phenotype. Is there another way of inducing calcium dysregulation that could possibly accelerate the phenotype? What about causing seizures in another way such as with PTZ? If there are no straightforward, good experiments that would help tease apart how this rescue is happening, then I do not think it is required for publication. The phenotype and rescue are exciting, however, and if there are a couple of additional tests that could be revealing, then I think it would be worth adding.

Reply: To induce seizures in zebrafish the most commonly used chemoconvulsant is PTZ (Review: D'amora et al. 2023, Int J Mol Sci.). There are other chemoconvulsants such as pilocarpine and Kainic acid. However their mechanism of action on the neurotransmitters are a little different; the end result is overexcitation of excitatory neurons. Therefore, we would not expect that these other drugs would give a different result from PTZ.

Third decision letter

MS ID#: dev.204348R2

MS TITLE: The non-canonical thioreductase *Tmx2b* is essential for neuronal survival during zebrafish embryonic brain development

AUTHORS: Jordy Dekker, Wendy Lam, Rachel Schot, Floris Ophorst, Herma C van der Linde, Leslie Sanderson, Gert-Jan Kremers, Woutje M Berdowski, Charlotte de Konink, Geeske M van Woerden, Grazia MS Mancini and Tjakko J van Ham
ARTICLE TYPE: Research Article

Dear Dr van Ham,

I am delighted to tell you that your manuscript has been accepted for publication in Development, pending our standard publication integrity checks.